# Single-cell trajectories reconstruction, exploration and mapping of omics data with STREAM

Huidong Chen [1,2,3,4], Luca Albergante [5,6,7], Jonathan Y. Hsu[1,8], Caleb A. Lareau [1,9], Giosuè Lo Bosco [10,11], Jihong Guan[4], Shuigeng Zhou[12], Alexander N. Gorban [13,14], Daniel E. Bauer[9,15], Martin J. Aryee [1,3,9], David M. Langenau[1,16], Andrei Zinovyev [5,6,7,14], Jason D. Buenrostro [9,17], Guo-Cheng Yuan [2,3,16] & Luca Pinello [1,9]

Single-cell transcriptomic assays have enabled the de novo reconstruction of lineage differentiation trajectories, along with the characterization of cellular heterogeneity and state transitions. Several methods have been developed for reconstructing developmental trajectories from single-cell transcriptomic data, but efforts on analyzing single-cell epigenomic data and on trajectory visualization remain limited. Here we present STREAM, an interactive pipeline capable of disentangling and visualizing complex branching trajectories from both single-cell transcriptomic and epigenomic data. We have tested STREAM on several synthetic and real datasets generated with different single-cell technologies. We further demonstrate its utility for understanding myoblast differentiation and disentangling known heterogeneity in hematopoiesis for different organisms. STREAM is an open-source software package.

[1] Molecular Pathology Unit & Cancer Center, Massachusetts General Hospital Research Institute and Harvard Medical School, Boston, MA 02114, USA. [2] Department of Biostatistics and Computational Biology, Dana-Farber Cancer Institute, Boston, MA 02215, USA. [3] Department of Biostatistics, Harvard T.H. Chan School of Public Health, Boston, MA 02215, USA. [4] Department of Computer Science and Technology, Tongji University, 201804 Shanghai, China. [5] Institut Curie, PSL Research University, F-75005 Paris, France. [6] INSERM, U900, F-75005 Paris, France. [7] MINES ParisTech, PSL Research University, CBIO-Centre for Computational Biology, F-75006 Paris, France. [8] Department of Biological Engineering, Massachusetts Institute of Technology, Cambridge, MA, USA. [9] Broad Institute of MIT and Harvard, Cambridge, MA 02142, USA. [10] Department of Mathematics and Computer Science, University of Palermo, 90123 Palermo, Italy. [11] Department of Sciences for technological innovation, Euro-Mediterranean Institute of Science and Technology, 90139 Palermo, Italy. [12] Shanghai Key Lab of Intelligent Information Processing, and School of Computer Science, Fudan University, 200433 Shanghai, China. [13] Department of Mathematics, University of Leicester, University Road, Leicester LE1 7RH, UK. [14] Lobachevsky University, Nizhni Novgorod 603022, Russia. [15] Division of Hematology/Oncology, Boston Children's Hospital, Department of Pediatric Oncology, Dana-Farber Cancer Institute, Department of Pediatrics, Harvard Medical School, Boston, MA 02215, USA. [16] Harvard Stem Cell Institute, Cambridge, MA 02138, USA. [17] Harvard Society of Fellows, Harvard University, Cambridge, MA 02138, USA. Correspondence and requests for materials should be addressed to G.-C.Y. (email: gcyuan@jimmy.harvard.edu) or to L.P. (email: lpinello@mgh.harvard.edu)

The rapid development of single-cell sequencing technologies has allowed to explore biological systems with unprecedented resolution. It is now possible to easily profile individual cells instead of cell populations, which advanced our fundamental understanding of the intrinsic cellular heterogeneity and dynamics. Single-cell sequencing protocols have been developed to measure different molecular layers, including transcriptomics[1–6], epigenomics[7–10], and proteomics[11–13]. The combination of these powerful measurements makes it possible to study important biological processes such as gene regulation on a multi-omics scale. Despite these technical breakthroughs several analytical and computational challenges exist due to the intrinsic characteristics of single-cell sequencing data, including cell-to-cell variation, sparsity of the data, biological and technical noise, and dropout events[14,15]. Several methods have been developed to detect distinct cell types and to identify rare cell subpopulations by clustering from single-cell gene expression data[16–19]. However, cellular processes, such as cell differentiation and cell maturation, are dynamic in nature and not always well described by discrete analysis like clustering. Therefore, other methods such as single-cell trajectory inference and pseudotime estimation have emerged. These methods allow to study cellular dynamics, delineate cell developmental lineages, and characterize the transition between different cell states. Briefly, single cells are ordered along deterministic[20–23] or probabilistic[24,25] trajectories and a numeric value referred to as pseudotime is assigned to each cell to indicate how far it progresses along a dynamic process of interest.

Although many computational methods have been developed for this task, these methods have been designed for analyzing single-cell transcriptomic data only. On the other hand, no end-to-end and open-source software solution exists, to our knowledge, to characterize epigenomic data such as single-cell chromatin-accessibility data (scATAC-seq). In addition, efforts on trajectory visualization remain limited. Current methods mainly focus on displaying single cells or clusters (or stable states) along the pseudotime, which makes it difficult to study subpopulation composition and its continuous transition along trajectories, especially for large datasets. Also, no trajectory inference method provides the possibility to map new cells to previously obtained reference trajectories without pooling cells and re-computing trajectories.

To fill these gaps, we have developed STREAM (Single-cell Trajectories Reconstruction, Exploration And Mapping), a comprehensive single-cell trajectory analysis pipeline, which can robustly reconstruct complex trajectories along with accurate pseudotime estimation from both single-cell transcriptomic data and chromatin-accessibility data. STREAM also provides a mapping feature and a set of interactive tools to explore and visualize both cell type composition and relevant genes (or transcription factor binding dynamics for scATAC-seq data) along the inferred trajectories.

## Results

**STREAM overview.** STREAM is a trajectory inference method that can accurately reconstruct complex developmental trajectories. It also provides informative and intuitive visualizations to recover and highlight important genes that define subpopulations and cell types. STREAM takes as input a single-cell gene expression (or epigenomic profile) matrix and approximates the data in three or more dimensions with a structure called the principal graph, a set of curves that naturally describe the cells' pseudotime, trajectories, and branching points (Fig. 1a). To reconstruct this structure, STREAM first identifies informative features such as variable genes or top principal components. Using these features, cells are then projected to a lower

dimensional space using a non-linear dimensionality reduction method called Modified Locally Linear Embedding (MLLE), which preserves distances within local neighborhoods. In the MLLE embedding, STREAM infers cellular trajectories using an Elastic Principal Graph implementation called ElPiGraph[26]. ElPiGraph is a completely redesigned algorithm for the previously introduced elastic principal graph optimization[27–29] based on the use of elastic matrix Laplacian, trimmed mean square error, explicit control of topological complexity and scalability to millions of points on an ordinary laptop. In STREAM, the ElPiGraph was further developed to integrate a new heuristic graph structure seeding to learn principal graphs in high dimensions with several problem-specific topological graph grammar rules optimized for single-cell trajectory inference (Methods, Supplementary Fig. 1).

To illustrate STREAM, we first reanalyzed a published scRNA-seq dataset from Nestorowa et al.[30]. In this study, 1656 single cells from the mouse hematopoietic system were sorted and profiled. Starting from the hematopoietic stem cells (HSCs), STREAM accurately recapitulates known bifurcation events in lymphoid, myeloid, and erythroid lineages and positions the multipotent progenitors before the first bifurcation event (Fig. 1b–d). To facilitate the exploration of the inferred structure, STREAM includes a flat tree plot that intuitively represents trajectories as linear segments on a 2D plane. In this representation, the lengths of tree branches are preserved from the MLLE embedding (Fig. 1b). In addition, cells are projected onto the tree according to their pseudotime locations and the distances from their assigned branches. If the process under study has a natural starting point (for example a known origin in a developmental hierarchy or a given sampling time point), the user can specify a root node. This allows easy re-organization of the tree using a breadth-first search to obtain a subway map plot that better represents pseudotime progression from a selected starting node (Fig. 1c). Although these visualizations capture trajectories and branching points, they are not informative on the density and composition of cell types along pseudotime, a common challenge when modeling large datasets. In fact, density information, an aspect overlooked by existing methods, is important to track not only how the composition of subpopulations changes along a trajectory but also how they get partitioned around branching events. To solve this problem, we develop a trajectory visualization method called the stream plot. This compact representation summarizes cellular developmental trajectories, user-defined annotations, branching points, cell density, and gene expression patterns (Fig. 1d). Additionally, STREAM detects potential marker genes of different types: diverging genes, i.e., genes important in defining branching points that are differentially expressed between diverging branches, and transition genes, i.e., genes for which the expression correlates with the cell pseudotime on a given branch. The expression patterns of the discovered genes can then be visualized using either subway map or stream plots (Fig. 1e, f, Supplementary Figs. 2–3, Supplementary Note 1).

**STREAM mapping procedure.** STREAM is the only trajectory inference method that explicitly implements a mapping procedure, which allows reusing a previously inferred principal graph as reference to map new cells not included in the original fitting procedure. Briefly, the STREAM mapping procedure maps new cells to the inferred structure using the neighbor relationships between new and old cells and the graph structure (see Methods). This can be accomplished since all the steps are deterministic and the MLLE dimensionality reduction provides an explicit function that maps points from the original space to the target subspace. A reference structure is important when studying genetic or epigenetic perturbation, or when comparing different conditions

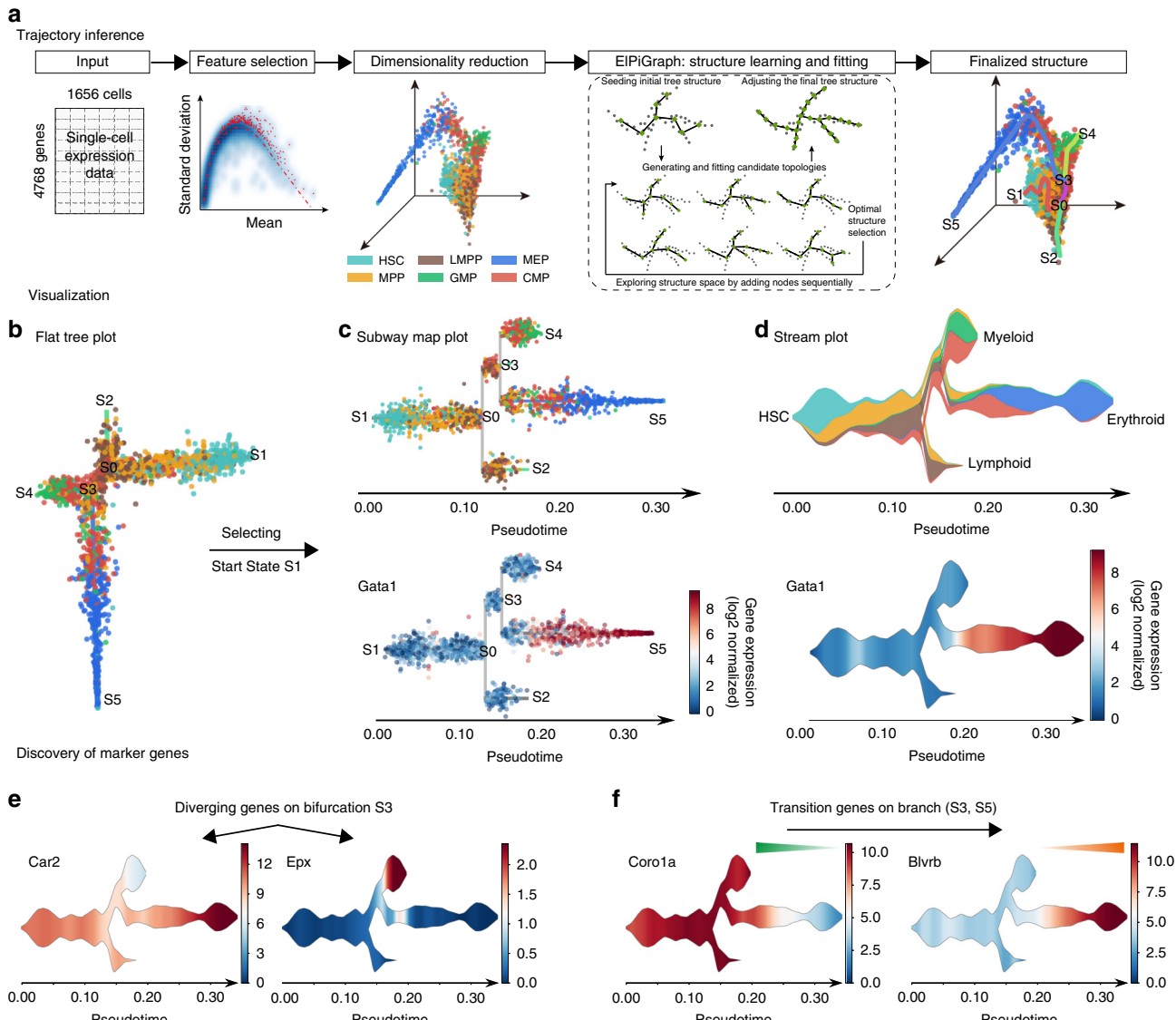

**Fig. 1** STREAM pipeline overview on single-cell RNA-seq data from the mouse hematopoietic system. **a** STREAM trajectory inference. Starting with a single-cell gene expression matrix, STREAM performs three main steps: selection of informative genes, dimensionality reduction, and simultaneous tree structure learning and fitting by ElPiGraph. The optimal structure is selected based on the elastic energy minimization among a set of candidate structures that are constructed every time a tree node is added. The final tree is interpreted as a set of connected curves representing different trajectories. **b**–**d** STREAM visualization of inferred branching points, trajectories, and expression of key genes at both single-cell level and density level. **b** Flat tree plot, branches are represented as straight lines and each circle represents a single-cell. The lengths of the branches and the distances between cells and their assigned branches are preserved from the space where trajectories were inferred. **c** Subway map plot, after selecting an initial state in the flat tree plot, the tree is re-ordered to facilitate visualization. Each cell is colored by a cell label, if provided (top), or based on the expression of a gene of interest (bottom). **d** Stream plot, an intuitive visualization to show cell density along different trajectories: at a given pseudotime, the width of each branch is proportional to the total number of cells (top). Stream plots can also visualize the expression of a gene of interest (bottom). **e**, **f** STREAM detection of marker genes. **e** STREAM automatically discovers important marker genes for each branch. Left, identification of differentially expressed genes between bifurcating branches. **f** Identification of transition genes (expression values correlate with pseudotime) along one specific branch. Top two detected differentially expressed genes (*Car2* and *Epx*) and transition genes (*Coro1a* and *Blvrb*) are shown, respectively, with stream plots

(for example normal and cancer, response to stimuli, etc.). In fact, the mapping procedure not only avoids pooling old and new cells and re-computing trajectories from scratch (a computationally-intensive operation), but more importantly does not distort the original structure. Keeping the original structure unperturbed is important to avoid incorrect interpretations of the reference pseudotime.

To illustrate the utility of the mapping feature, we applied STREAM to analyze a published scRNA-seq dataset from Olsson et al.[31]. This study focused on the mouse hematopoietic system,

specifically on the consequences of cell-fate determination within the granulocyte monocyte progenitors (GMP) population after the knockout of important master regulators. Using FACS sorting, 382 cells were isolated and profiled from different subpopulations, including stem/multipotent progenitor (LSK; lin−, Sca1+, c-Kit+), CMP, GMP, and LKCD34+ (lin−c-Kit + CD34+) cells (Fig. 2a left). A key result of this study is the discovery of metastable mixed-lineage states and the presence of co-expressed genes at single-cell level from competing lineages. The authors suggest that these metastable states are important in

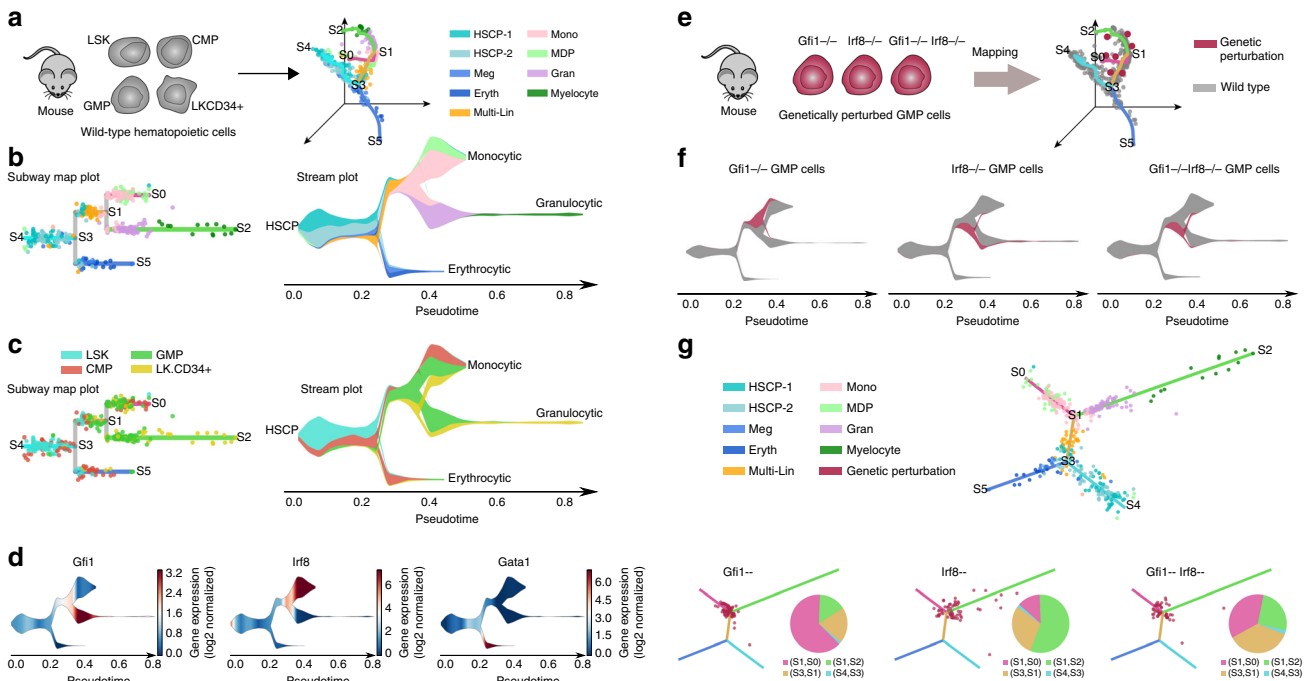

**Fig. 2** Mapping of genetic perturbation data to the inferred trajectories from wild-type mouse hematopoietic cells. **a** Left, cell subpopulations were isolated from the mouse wild-type hematopoietic system including stem/multipotent progenitor (LSK), common myeloid progenitor (CMP), granulocyte monocyte progenitor (GMP), and LKCD34+ and profiled by scRNA-seq. Right, wild-type cells are shown in low dimensional space together with trajectories inferred by STREAM. Cells are colored by the cluster labels proposed by Olsson at al. **b** Left, subway map plot; right, stream plot. Both are colored by cluster labels inferred by Olsson at al. **c** The same subway map and stream plot as **b** but colored by FACS gating labels from the original study. **d** Stream plots of three key marker genes: Gfi1 for granulocyte, Irf8 for monocyte and Gata1 for Meg and Eryth. **e** Left, scRNA-seq is performed on genetically perturbed cells within the GMP populations: Gfi1−/−, Irf8−/−, and Gfi1−/−Irf8−/−. Right, genetically perturbed cells are mapped using STREAM to the low dimensional space in which cellular trajectories were built based on wild-type cells. **f** At density level, stream plots easily summarize the effects of the three genetic perturbations: Gfi1−/− cells are diverted to monocyte-committed branch while Irf8−/− cells are instead diverted to granulocyte-committed branch. Gfi1−/−Irf8−/− cells have equal chances to differentiate into either branch. **g** Single-cell level visualization of perturbed cells on the reference flat tree plot constructed from wild-type cells (top). Genetically perturbed cells are mapped to the flat tree and shown in red. Pie charts show the proportion of genetically perturbed cells on different branches. Consistently with the stream plot in **e**, Gfi1−/− cells mainly appear on monocyte-committed branch (S1, S0), while the majority of Irf8−/− cells appear on granulocyte-committed branch (S1, S2). Gfi1−/−Irf8−/− cells are approximately equally located on the intermediate state branch (S3, S1), monocyte-committed branch (S1, S0), and granulocyte-committed branch (S1, S2)

cell-fate decisions and that master regulators play a key role in this process. In fact, they uncovered and validated two key transcription factors, i.e., Gfi1 and Irf8, that are co-expressed in a subpopulation and are shown to be important for the commitment to neutrophils or macrophages. Importantly, this dataset contains, in addition to wild-type data, genetic perturbations of those two key regulators.

Using the wild-type data, STREAM unbiasedly and correctly reconstructed the cell lineage hierarchy as shown by inspection of the labels proposed in the original study (either cell surface markers or predicted lineages) (Fig. 2a right). Starting from hematopoietic stem cell/progenitor (HSCP), cell lineage bifurcates into an erythrocytic branch (which contains megakaryocytic (Meg) and erythrocytic (Eryth) cells) and into a multi-lineage primed (Multi-Lin) branch. Multi-Lin cell lineage further separates into the granulocytic (Gran) branch and monocytic (Mono) branch. The hierarchical progression can be easily visualized by our proposed 2D visualizations: subway map and stream plots (Fig. 2b). Importantly, STREAM precisely recovers the bifurcation event from Multi-lineage to Mono and Gran as shown in the original study within the wild-type GMP cellular population (Fig. 2b, c), whereas the proposed Monocle2 analysis of the same dataset[20] incorrectly assigns Multi-Lin cells to a very short erythroid branch. Furthermore, Monocle2 branch lengths are overall very diverse and distorted in their hierarchical

representation (F_E branch in Fig. 2, and Supplementary Fig. 18 of the original paper[20]). Based on our analysis, the Gran-specific gene Gfi1, Mono-specific gene Irf8, and Eryth-specific gene Gata1 are highly expressed on their respective inferred trajectories, confirming the validity of the reconstructed branching structure (Fig. 2d).

Next, using the STREAM mapping function, we analyzed the genetic perturbation data to study the consequences on cell-fate determination of Gfi1 loss (Gfi1−/−), Irf8 loss (Irf8−/−) and both Gfi1 and Irf8 loss (Gfi1−/− Irf8−/−) within wild-type GMP cells (Fig. 2e). Gfi1−/− GMP cells tend to differentiate into the Mono branch and Irf8−/− GMP cells lean toward the Gran branch. The combined loss of Gfi1 and Irf8 instead does not show any imbalance of cells differentiating into the diverging branches (Fig. 2f, g). Our predictions are validated by the original study where the authors used GMP cells with inducible expression and GFP reporters for Gfi1 and Irf8. Irf8 loss led to cells that differentiated toward granulocyte. Conversely, Gfi1 loss led the cells to differentiate toward monocytes. Interestingly they showed that cells from the hematopoietic stem cell/progenitor and myeloid compartments are trapped with the double knockouts of Irf8 and Gfi1, and in fact, are rarely differentiating towards monocytes or granulocytes. These results are in full agreement with our unbiased analysis. In addition, compared to the Monocle2 analysis of this dataset, our reference structure can

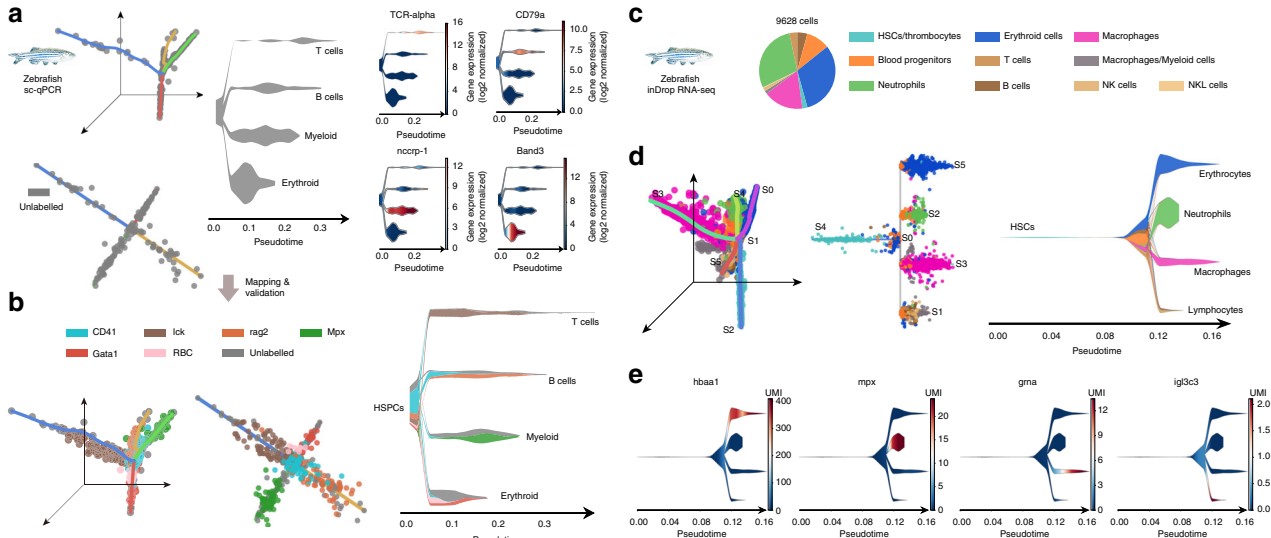

**Fig. 3** STREAM recovers developmental trajectories of hematopoietic cells in zebrafish from qPCR and inDrop data. **a** STREAM output for single-cell qPCR on cells from zebrafish wild-type whole-kidney marrow (WKM). The stream plot shows only one color (gray) since no labels to annotate cell-types are available in this case. Four trajectories are recovered and visualized in the 3D space, flat tree plot and stream plot respectively. Four of the top marker genes automatically detected by STREAM are visualized as stream plots: *TCR-alpha* (T cells), *cd79* (B cell), *nccrp-1*(myeloid), and *band3*(erythroid). **b** Validation of the putative cellular differentiation branches. Hematopoietic cells from adult transgenic zebrafish and peripheral blood are mapped to the trajectories inferred in **a**. These cells comprise peripheral red blood cells (RBC) and FACS-sorted cells, which include *CD41* (hematopoietic stem and precursor cells), *lck* (T cells), *Mpx* (myeloid cells), *rag2*(B cells), and *Gata1* (erythroid cells). **c** STREAM output for inDrop single-cell RNA-seq data from the zebrafish wild-type whole-kidney marrow. Cell labels are based on the Tang et al. classification and are highly unbalanced as shown by the pie chart. **d** Principal graph plot, subway map plot and stream plot show the trajectories recovered in the hematopoiesis of zebrafish. HSCs through blood progenitor cells differentiate into erythroid, myeloid (including neutrophil and macrophage) and lymphoid cells. **e** Marker genes from the original study or automatically detected are visualized using stream plots to confirm and validate the recovered structure

be fixed to recapitulate only the wild-type cells and is not influenced by the fmapping of new cells, whereas the Monocle2 analysis requires to recompute trajectories (compare A,B with C, D in Supplementary Fig. 18 of the original paper[20]).

Taken together, STREAM recovers the correct trajectories for the wild-type cells and, using the mapping feature, also predicts and effectively visualizes the consequences of the genetic perturbation as validated in the original study.

**Delineating hierarchies of the zebrafish hematopoiesis**. To test the robustness and scalability of STREAM, we next explored data derived from different platforms and organisms. To this end, we used two recently published zebrafish datasets obtained with single-cell qPCR[32] and inDrop[33] assays. These data provided the first comprehensive model of the zebrafish hematopoiesis system without biases introduced by FACS sorting subpopulations.

The first dataset from Moore et al.[32] provided a first model of the zebrafish hematopoiesis system using a carefully designed panel of 96 genes. In this study, 166 cells were profiled from the wild-type (WT) whole-kidney marrow (WKM). STREAM analysis uncovered four cell lineages trajectories (Fig. 3a left) and based on the automatic gene detection module, uncovered marker genes for each trajectory (Fig. 3a right), which includes T cell marker gene *TCR-alpha*, B-cell marker gene *CD79a*, myeloid marker gene *nccrp-1*, and erythroid marker gene *band3*. Based on this analysis, we hypothesize that the inferred four branches correspond to T cell, B cell, myeloid, and erythroid lineages (Fig. 3a middle). To test this hypothesis, we used the STREAM mapping feature to map fluorescent-labeled and FACS-sorted cells from WKM: 20 erythroid cells from peripheral blood cells (per RBC), 24 erythroid cells Tg(gata1:dsRed), 48 myeloid cells Tg(mpx:GFP), 49 marrow-derived B cells Tg (rag2:dsRed), 83

mature T cells Tg (lck:GFP)cells, 85 HSPCs Tg(CD41:GFP)low. lck + cells were mapped to the T cell branch, mpx + cells were mapped to the myeloid cell branch, rag2 + cells were mapped to the B cell branch, both gata1+ and peripheral RBCs were mapped to the erythroid branch, while the majority of HSPCs were mapped to the proposed starting state as expected (Fig. 3b). This result provides additional support for our hypothesis that the four branches corresponded to well defined lineages, highlighting the utility of the STREAM mapping feature.

To test the scalability and robustness of STREAM on a larger and more challenging scRNA-seq dataset, we next analyzed 9628 unlabeled cells from the zebrafish whole-kidney marrow generated by Tang et al.[33] using the inDrop protocol[2]. The original study, based on dimensionality reduction and clustering, uncovered and annotated 10 different and imbalanced subpopulations (some of which were validated by the authors using sorting of fluorescent transgenic cell sub-populations) (Fig. 3c). STREAM correctly recapitulated the hierarchy of the different lineages and unbiasedly recovered four main hematopoietic cellular trajectories: starting from HSCs, through blood progenitor cells, cells differentiate into erythroid, macrophage, neutrophil, and lymphoid lineages (Fig. 3d). Importantly, we rediscovered well-known marker genes: *hbaa1* for the erythroid branch, *grna* for the macrophage branch, *mpx* for the neutrophil branch, and *igl3c3* for the lymphoid branch (Fig. 3e).

However, we noticed that B and T cells were not separated and were assigned to the same lineage branch. Therefore, we derived an improved seeding strategy that is well suited to learn complex trajectories in high dimensions and that well recapitulates the known lineage for this dataset as presented in Supplementary Note 2 and Supplementary Figs. 4–6. This new strategy is generalizable to other datasets and described in detail in the method section.

In summary, these analyses highlight some important points of our approach: (1) STREAM is able to identify more refined trajectories increasing the number of dimensions, (2) we can recover trajectories using unsorted populations, (3) the trajectory inference is robust to subpopulation imbalance, (4) our gene expression analysis is a powerful tool to discover marker genes, and (5) our method is scalable to currently available large-scale single-cell assays.

**Comparison with other methods**. Several methods have been proposed for pseudotime inference or trajectory reconstructions. In fact, more than 50 methods have been proposed for this task, making a systematic comparison unfeasible for the scope of this manuscript. For this reason, we compared STREAM with 10 state-of-the-art methods well recognized and commonly used by the single-cell community: Monocle2, scTDA, Wishbone, TSCAN, SLICER, DPT, GPFates, Mpath, SCUBA, and PHATE[20–24,34–38]. An overall summary of these different methods, including their general features, required inputs, supported assays, scalability, and execution time, can be found in Supplementary Table 1 and Supplementary Table 2, and a short discussion about the core algorithms used by each method is presented in Supplementary Note 3.

In our quantitative comparison we focused on two important aspects: topology correctness and pseudotime accuracy. We also present in our assessment the default visualizations provided by each method to showcase and easily compare their expressiveness in representing cellular development trajectories. For each method, the analyses were performed with standard parameters when possible (following the guidelines provided in the documentation) otherwise the parameters were obtained by contacting the respective authors (Supplementary Note 3).

To evaluate the ability of each method in recovering the correct topology we used a proposed gold-standard synthetic dataset by Rizvi et al.[34] with known topology and pseudotime: two bifurcation events and three different time points (Fig. 4a).

First, we started by quantitatively evaluating the number of correctly detected branching events and the pseudotime accuracy (Online methods). For topology correctness, STREAM and other five methods, including scTDA, SLICER, Monocle2, Mpath, GPfates, PHATE, successfully identified two bifurcation events. Second, for pseudotime accuracy, we calculated for each method the correlation between true pseudotime and inferred pseudotime as proposed before[34]. Four metrics were used to evaluate this correlation, including rank-based Pearson correlation, distance-based Pearson correlation, Spearman's rank correlation, and Kendall's tau coefficient. We use four different correlation metrics since some methods (scTDA, TSCAN, and Mpath) only return a simple ordering, i.e., the ranks of cells, and do not provide the actual pseudotime defined as the distance of each cell from the origin in the proposed embedding (for Spearman's rank correlation and Kendall's tau coefficient, ranks-based and distance-based correlations are the same). STREAM has the best performance for two out of four metrics (and importantly when using distance-based pseudotime) and second-best performance for the other two rank-based metrics (following scTDA in which this synthetic dataset was proposed) (Fig. 4b). Finally, we assessed the qualitative output of each method using their proposed visualization. STREAM is the only tool that provides a density-level visualization to study the composition of different cell types in different branches. (Fig. 4c, Supplementary Note 3).

To compare the different methods on real datasets, we first used one of the most commonly used scRNA-seq datasets for this task, originally generated by Trapnell et al.[39]. This dataset contains human skeletal muscle myoblasts (HSMM) cells

differentiating along a linear trajectory. In this analysis, we were able to evaluate only methods capable to detect the correct bifurcation event. Regardless, the visual outputs of all the methods are presented for completeness (Supplementary Fig. 7). The original study proposed a single bifurcation, which leads to myoblast cells or separate potentially contaminating cells (Fig. 5a). To test the quality of pseudotime it has been proposed to correlate known marker gene expression (a surrogate for the correct ordering) along the myoblast differentiation trajectory with the rank or distance-based pseudotime (Online Methods). To this end, we used the previously proposed genes *ENO3*, *MEF2C*, and *MYH2*[20,35]. When ordering cells by pseudotime, we expect a monotonic increase of the marker gene expression. Importantly, when ordering cells by distance-based pseudotime, we expect, in the ideal scenario, a continuous and smooth distribution. For example, STREAM generates a smooth and monotonically increasing distribution of *ENO3* expression based on the inferred pseudotime as shown in (Fig. 5b). In contrast, we noticed that for the distance-based pseudotime, in Monocle2, cells are mainly attracted to the end points of the trajectories, with few cells in between (Supplementary Fig. 8). In Wishbone and SLICER, distance-based pseudotime shows a set of unexpected discrete segments. Neither Mpath nor TSCAN can generate distance-based pseudotime. In addition, Mpath does not recover a monotonically increasing trend (Supplementary Fig. 8). STREAM has also the highest average coefficient on *ENO3* based on the four different metrics (Fig. 5c). When combining all three marker genes, STREAM has the overall best performance (calculated as the average rank for the four proposed metrics) (Fig. 5d).

Finally, we analyzed a high-quality single-cell qPCR dataset containing ~270 blood cells sorted from six different populations: HSC, MPP, CMP, GMP, MEP, and common lymphoid progenitor cells (CLPs) profiled for ~170 key transcription factors important in mouse hematopoiesis[40]. The output of each method is shown in Supplementary Fig. 9. STREAM is the only method that clearly shows the reconstructed developmental trajectories and the lineage hierarchies using its default visualizations. STREAM recovers a trajectory that starts from HSCs and then through MPPs bifurcates into CMPs and a subset of likely erythroid-poised CMPs shows an early progression into MEP, consistent with a recently refined model of hematopoiesis[41]. STREAM recovers also a second bifurcation event that effectively captures cell commitment from MPPs into GMPs and CLPs

To assess the quality of the discovered trajectories, we reasoned that classic marker genes for different lineages should be expressed in cells belonging to different trajectories with minimal mixing (i.e., it should be rare to observe single cells that express simultaneously both markers). To this end, we selected *Gata1*, a classic erythroid marker, and *Pax5*, a classic lymphoid marker. For each method, we selected the two best branches that contained the most *Gata1* or *Pax5*-expressing cells, respectively. Then, each branch is evaluated based on precision, recall and the F1 score (Methods). The optimal model should balance precision and recall separating *Gata1* and *Pax5* in two distinct branches; whereas under-branching models will have a high recall, but poor precision and over-branching models will have a high precision but poor recall (Fig. 5e).

STREAM has the highest F1 score for both *Gata1* and *Pax5* among all the methods tested and balance well precision and recall (Fig. 5f, g). SCUBA works reasonably well for both genes but has a lower recall overall. Monocle2 tends to generate over-branching structures with high precision but poor recall. Mpath works well in the case of *Gata1* but performs poorly for *Pax5*. Wishbone and DPT have relatively low precision scores because part of cells expressing *Gata1* are misplaced on the *Pax5*-expressing branch.

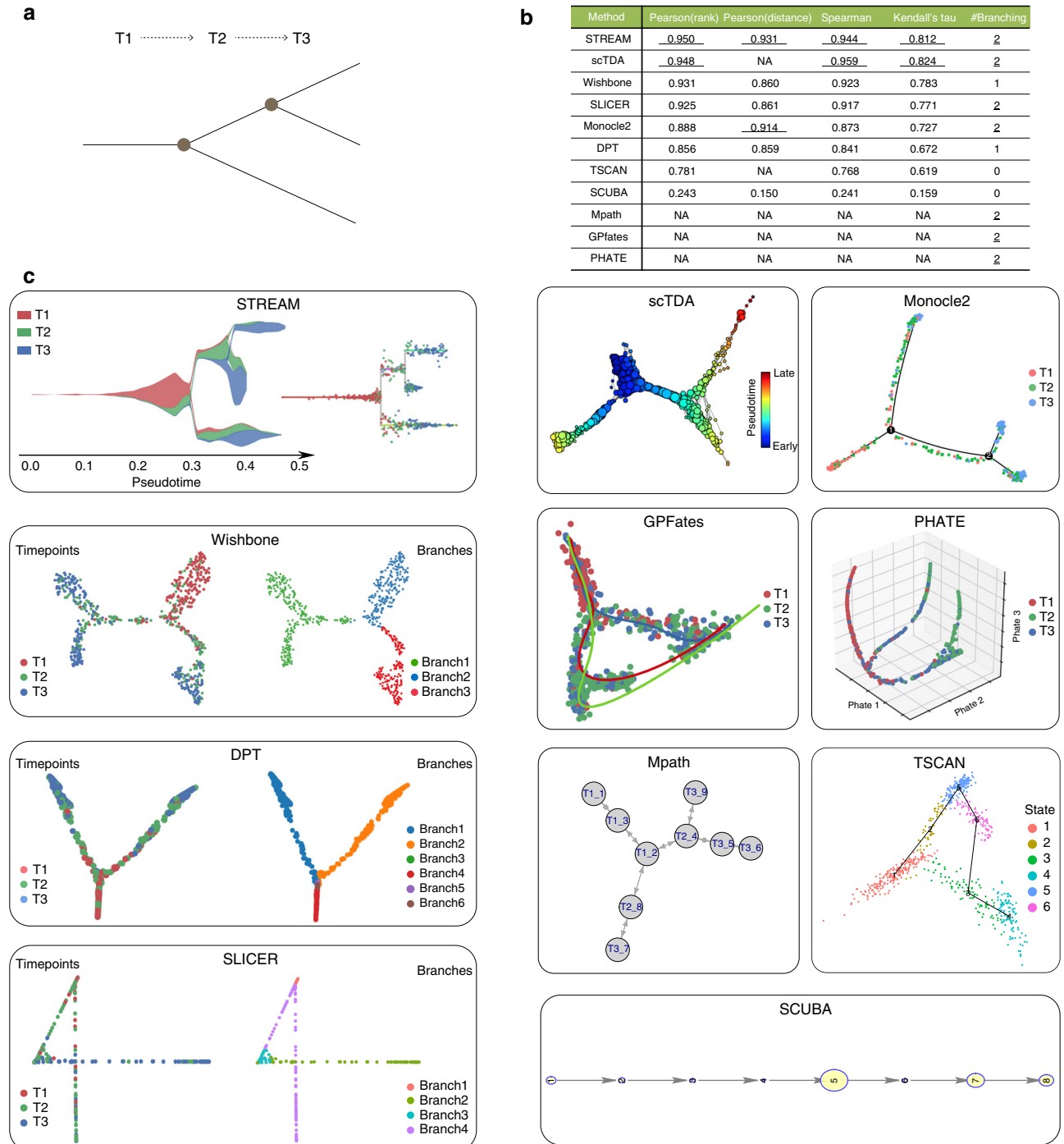

**Fig. 4** Comparison with other methods on synthetic datasets. **a** Topology structure of a synthetic benchmark dataset proposed by Rizvi et al. Cells are sampled from three different time points (T1, T2, and T3). It has two bifurcation events: the first happens between T1 and T2, the second happens between T2 and T3. **b** Correlation (Pearson, Spearman and Kendall's tau) between real and inferred pseudotime (either rank-based or distance-based pseudotime) and the number of correctly detected branching events for all eleven methods tested. **c** Output of different trajectories inference methods. STREAM, stream plot and subway map plot allow to study cellular trajectories and time points at both density level and single-cell level. Wishbone, left, cells are colored by time points, right, cells are colored by its identified differentiation branch ID. scTDA, proposed topological representation colored by pseudotime indicates cellular differentiation trajectories, in which nodes correspond to a set of cells and node size is proportional to the number of cells. Monocle2, cells are colored by time points and the skeleton depicts differentiation trajectories. DPT, on the left cells are colored by time points and on the right cells are colored by its identified differentiation branch ID. GPFates, cells are colored by time points. Three curves represent three trajectories(trends). PHATE, cells are colored by time points. SLICER, on the left cells are colored by time points and on the right cells are colored by its identified differentiation branch ID. Mpath, each node represents one landmark cell and the tree structure shows differentiation trajectories. TSCAN, cells are colored by states detected from TSCAN. The skeleton depicts a linear trajectory. SCUBA, each node is one cluster and the node size represents the variance within each cluster

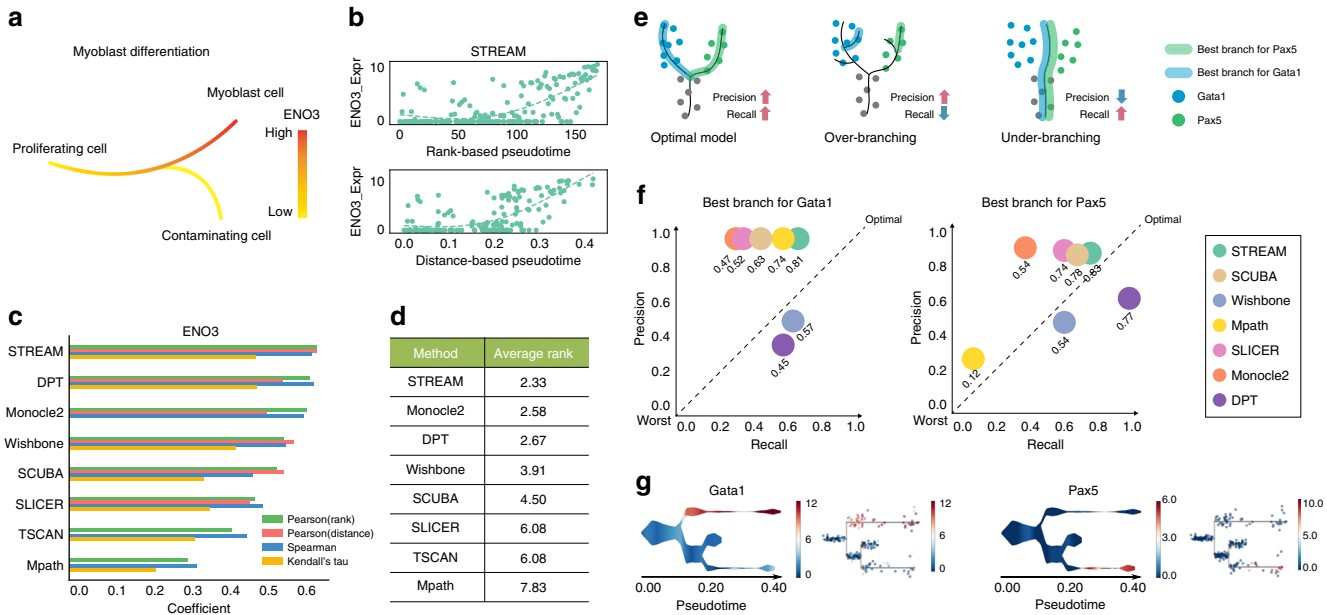

**Fig. 5** Comparison with other methods on real datasets. **a** *ENO3*, a marker gene for late-stage differentiation, is used to evaluate the myoblast commitment trajectory (highly expressed in fully differentiated cells). **b** Along the myoblast differentiation trajectory, the scatter plots show the relationship between *ENO3* expression and rank-based pseudotime or distance-based pseudotime inferred by STREAM. The dashed lines are fitted curves by generalized additive model (GAM). **c** Correlation analysis along myoblast differentiation. The bar plot shows for each method the Pearson correlation between inferred rank-based pseudotime and *ENO3* expression, Pearson correlation between inferred distance-based pseudotime and *ENO3* expression, Spearman correlation between inferred pseudotime, and *ENO3* expression and Kendall's tau correlation between inferred pseudotime and *ENO3* expression. Only methods that successfully detect the correct bifurcation are included. **d** The average rank of three maker genes, i.e., *ENO3*, *MYH2*, and *MEF2C* on four different correlation analysis metrics as described in **c**. **e** Using the two marker genes *Gata1* and *Pax5*, which are highly expressed on MEP-committed branch and CLP-committed branch, respectively, and rarely co-expresses in the same single cell, the illustration shows three different reconstruction scenarios: optimal model (high precision and high recall), over-branching (high precision but low recall) and under-branching (low precision but high recall). **f** The scatter plots show the relationship between precision and recall for each marker gene across methods. Each circle represents one method and its F₁ score is reported below. **g** Stream and subway map plots of the mutually exclusive genes *Gata1* and *Pax5*

In summary, although many of the existing methods work reasonably well with simple linear trajectories, they may provide over- or under-branched models in more complex scenarios and may mask important trajectories or marker genes.

**STREAM reconstructs trajectories from scATAC-seq data.** In addition to single-cell transcriptomic profiling, novel assays have been proposed to capture chromatin-accessibility at single-cell resolution[7,10]. These assays allow to study an important additional layer in gene regulation. In fact, accessible sites in the genome can be used to profile the activity of important cell type specific non-coding regulatory elements such as enhancers. These regions harbor transcription factor binding sites and can control the activity of relatively proximal genes via long-range interactions[7].

The analysis of human scATAC-seq data is particularly challenging because sparsity is intrinsic to these assays. In fact, the signal is limited by the DNA copy number, which only consists of 0, 1, or 2 reads within a diploid genome. In addition, compared to previously published datasets in other model organisms (mouse, *Drosophila melanogaster*), the human genome is larger (respectively, 1.2X and 27X) making this problem even more accentuated. Although some initial efforts have been made to adapt existing trajectory inference methods originally developed for transcriptomic data to scATAC-seq data analysis[42,43], to our knowledge STREAM is the only documented end-to-end pipeline that provides to users the specific functions to analyze and visualize scATAC-seq data starting from raw count data and based on an unbiased approach to model important DNA

sequence features associated with chromatin-accessibility. We present below how STREAM can be used to infer trajectories from single-cell epigenomic data and show its application to a recently published dataset, where a total of 3072 cells were profiled from the human bone marrow and isolated by FACS into nine different cellular populations, including HSC, MPP, CMP, CLP, LMPP, GMP, MEP, mono and plasmacytoid dendritic cells (pDCs)[44].

In STREAM to overcome the sparsity of the data and the limited ability to capture a given region in a single cell we focus on chromatin-accessibility variable regions across cells instead of scoring the entire genome (from $3.3 \times 10^9$ potential base pairs in the human genome we reduce it to ~450,000 regions covering only ~7%), and then aggregate over general features related to chromatin accessibility on these regions. To this end STREAM uses an unbiased set of DNA sequence features i.e., *k*-mers (word of length k on the DNA alphabet) and chromVAR[45] to calculate accessibility deviations across cells (Fig. 6a). Briefly, starting from count data, we construct a matrix of cells x *k*-mer accessibility z-scores (in our experiments $k = 7$). The *k*-mer accessibility z-scores can be used by STREAM as features to reconstruct trajectories. However, we observed that selecting the top principal components on the scaled z-score matrix allows filtering out potential small fluctuations further reduces the dimensionality and improves the quality of the recovered structures (Methods). This general and unbiased *k*-mer strategy is agnostic to any known transcription factor motifs and thus generalizable to other systems.

After filtering cells as previously described[44], single-cell accessibility profiles for 2034 high-quality cells passed quality

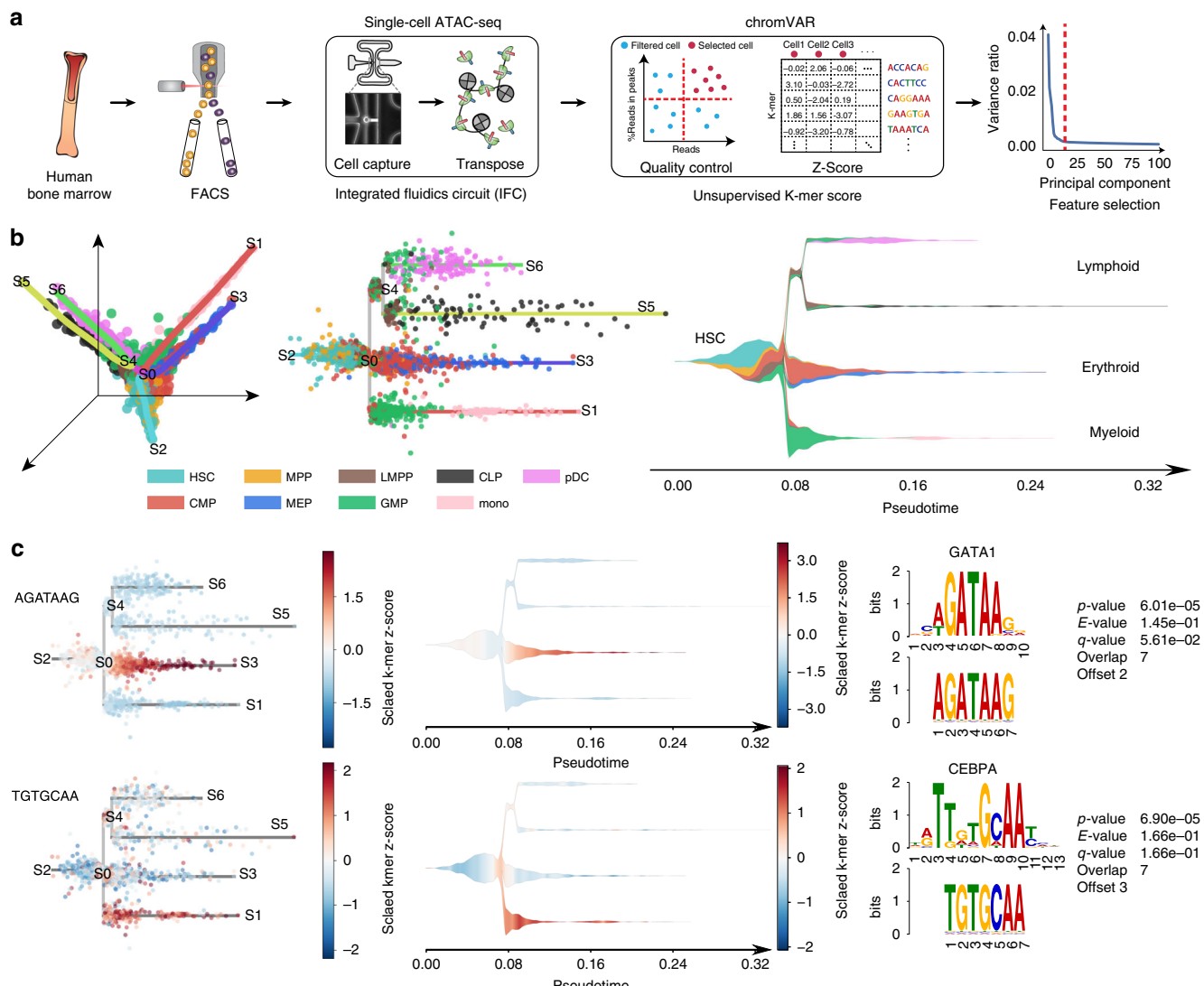

**Fig. 6** STREAM on single-cell epigenomic data from the human hematopoietic system. **a** Single-cell ATAC-seq workflow. FACS sorting is used to isolate populations from CD34+ human bone marrow and single-cell ATAC-seq measurements are performed. After mapping reads to the reference genome, reads within peaks are selected and ChromVAR is used to calculate $k$-mers z-scores. Finally, PCA is applied to the z-score matrix and top principal components are selected as features for the STREAM analysis. **b** STREAM learns a principal graph from chromatin-accessibility data and accurately reconstructs cellular developmental trajectories of the human hematopoiesis. As in Fig. 1, the structure can be easily visualized thanks to the subway map and stream plots. In the first branch, the HSCs segregate through MPP into lymphocyte-committed, erythrocyte-committed and myelocyte-committed branches. STREAM also reconstructs the bifurcation from lymphoid multipotent progenitors (LMPP) to CLP and plasmacytoid dendritic cells (pDC). **c** Discovery of transcription factors important for lineage commitment. 7-mer DNA sequences are automatically detected and their frequencies are visualized in both the subway map and stream plots. These sequences are mapped to known transcription factors motifs. We recovered *GATA1* and *CEBPA* as top hits, two classic master regulators in blood development, which correlate with directionality toward erythroid differentiation and myeloid differentiation, respectively

control. We emphasize that each cell was sorted using multiple surface markers as previously described[44], providing a phenotypic "true positive" for cell state that would enable us to determine the accuracy of STREAM.

STREAM not only accurately reconstructs cellular developmental trajectories of the human blood system, but also recovers key sequence features and master regulators that have been implicated in differentiation and lineage commitment for different subpopulations (Fig. 6b, c). In the inferred principal graph, the HSCs branch segregates through MPP into the erythrocyte-committed, lymphocyte-committed, and myelocyte-committed branches. STREAM also reconstructed the bifurcation from lymphoid multipotent progenitors (LMPP) to CLP and

plasmacytoid dendritic cells (pDC). Interestingly, STREAM reveals a similar and consistent hematopoietic hierarchy described by orthogonal assays such as transcriptomic profiling (Fig. 6b).

STREAM also uncovers annotated (i.e., mappable to transcription factors) or unannotated DNA sequences that may be important in defining the different developmental paths. In fact, using the inferred structure, STREAM automatically identifies significant $k$-mer DNA sequences for each branch. Importantly, those recovered $k$-mers can be mapped to known transcription factors motifs that may drive cell-fate decision and commitment. We uncovered *GATA1* and *CEPBA*, for erythroid lineage and myeloid lineages, respectively (Fig. 6c). We also uncovered several

additional potential regulators for HSCs (*ERG*[44], *HOXB8*[46]) and MEPs (*LMO2*[47], *TAL1*[44]) (Supplementary Fig. 10).

In summary, compared to previous studies, STREAM provides an unbiased reconstruction of human hematopoiesis using chromatin-accessibility data at single-cell resolution. STREAM is able to perform pseudotime ordering on human cell chromatin-accessibility data without relying on accessibility of known transcription factor binding sites[7] or a priori knowledge of sampling time[42], hence providing an unbiased approach.

## Discussion

Large-scale single-cell measurements have opened unprecedented opportunities to study dynamic processes such as differentiation or response to stimuli. Trajectory inference methods are important tools to describe those processes based on snapshot measurements.

In this study we presented STREAM, a trajectory inference tool capable of reliably reconstructing trajectories and inferring pseudotime from different data types and when multiple branching points are present. Our method does not require prior knowledge such as time points, start cell, or the number of branching events to reconstruct trajectories, and does not need extensive bioinformatics knowledge thanks to a user-friendly and interactive web interface. Additionally STREAM introduces four innovations compared to other existing methods: (1) a density-level trajectory visualization useful to study subpopulation composition and cell-fate genes along branching trajectories, (2) a documented end-to-end pipeline to reconstruct trajectories from chromatin-accessibility data, (3) an interactive website that can be used to not only compute trajectories but also host a database to readily visualize and explore precomputed trajectories for several published studies[30–33,39,44], and (4) a trajectory mapping procedure to readily map new cells to precomputed structures without pooling data and re-computing trajectories. This last innovation allows facile analysis of data from genetic perturbation studies or to assign diseased/stimulated cells to a normal/resting developmental hierarchy. Existing methods instead require fitting a new model since the fitting procedure is not deterministic or this feature is absent. The main problem with re-computing the structure lies in the fact that it is hard to interpret pseudotime and cell positioning since trajectories may change based on the density and/or composition of the new cells to map. Our mapping procedure is instead deterministic and allows the user to easily and predict perturbation effects and explore the origin of unknown cell populations on annotated branching structures or vice versa (see an example in Fig. 3). However, the mapping procedure may not be optimal if the new cells have totally different fates compared to the ones present in the reference structure as discussed in the Methods section.

STREAM has been extensively tested using several published datasets from different organisms (zebrafish, mouse, human) and single-cell technologies (qPCR, scRNA-seq, scATAC-seq). Our comparison of STREAM with 10 other methods shows that it is among the top scoring methods on both synthetic and real datasets (best average ranking across several metrics and most balanced method for branching detection). However, there are still several general limitations for the current trajectory inference methods, including STREAM. First, most of the methods are still based on linear or tree-structure models even though we know that developmental processes may involve more sophisticated topologies, e.g., cyclic or disconnected graph. Therefore, further improvement is necessary to adapt STREAM to properly describing these more general structures. Although ElPiGraph, the core algorithm behind STREAM, inherently supports learning complex graphs, more work is necessary to provide intuitive

visualizations to capture and represent more sophisticated topologies. Second, recent single-cell assays are now capable of profiling millions of cells; however, many of the current trajectory inference methods don't scale or have been tested only on a few thousand cells (see Supplementary Table 2). Last but not least, while integrating multi-omics is crucial to accurately describe the cellular developmental landscape, so far very few trajectory inference methods (Monocle2 and STREAM) have been applied to analyze single-cell chromatin accessibility. Therefore, we expect further development in this direction and on methods to properly integrate the different trajectories obtained from different omics data. In fact, in addition to single-cell measurement of gene expression and chromatin accessibility, single-cell assays have been proposed to measure other epigenetic information (such as single-cell methylation[48] or nucleosome positioning[49]) or other molecular measurements such as protein levels[11–13]. We believe that further developments of STREAM to support these additional assays are important and will require defining appropriate informative features together with distance or similarity metrics to properly compare cells and appropriately define pseudotime. Regardless of these challenges we believe the visualization procedures proposed in this study can be easily adopted irrespective of the particular inference procedure and the omics data used to infer the pseudotime.

Taken together, STREAM is an important tool to study cellular development and differentiation: it can accurately recover and describe complex developmental trajectories, it provides informative and intuitive visualizations to highlight important genes that define cell-fate decisions and subpopulation composition, and it is an accessible tool that enables researchers even with limited computational skills to analyze, explore and share their single-cell-based trajectory analyses and insights.

## Methods

**Feature selection.** For transcriptomic data (single-cell RNA-seq or qPCR), the input of STREAM is a gene expression matrix, where rows represent genes, columns represent cells. Each entry contains an adjusted gene expression value (library size normalization and log2 transformation). The most variable genes are selected as features, using a procedure we have previously proposed[50]. Briefly, for each gene, its mean value and standard deviation are calculated across all the cells. Then a non-parametric local regression method (LOESS) is used to fit the relationship between mean and standard deviation values. Genes above the curve that diverge significantly are selected as variable genes.

**Dimensionality reduction.** Each cell can be thought as a vector in a multi-dimensional vector space in which each component is the expression level of a gene. Typically, even after feature selection, each cell still has hundreds of components, making it difficult to reliably assess similarity or distances between cells, a problem often referred as the curse of dimensionality[51]. To mitigate this problem, starting from the genes selected in the previous step we project cells to a lower dimensional space using a non-linear dimensionality reduction method called Modified Locally Linear Embedding (MLLE)[52].

LLE-based methods generate a compact and continuous embedded structure that considers the local similarity of each cell with its neighbors. For standard LLE, each point $x_i, i = 1, ..., N$, in $R^m$ is reconstructed from its selected $k_i$ neighbors $\{x_j, j \in J_i\}$. The optimal single weight vector $W_i = \{w_{ji}, j \in J_i\}$ is determined by solving the constrained least squares problem:

$$min\left\|x_i - \sum_{j \in J_i} w_{ji}x_j\right\|, s.t. \sum_{j \in J_i} w_{ji} = 1 \tag{1}$$

Once the reconstruction weight vector $W_i$ is computed, LLE maps the data points $\{x_1, ..., x_N\}$ to vectors $T = \{t_1, ..., t_N\}$ in the lower dimensional space $R^d$ $(d < m)$ by minimizing the embedding cost function:

$$min\left\|t_i - \sum_{j \in J_i} w_{ji}t_j\right\|^2, s.t. TT^T = I \tag{2}$$

But using a single weight factor for each data point can result in an instability of the LLE procedure because of the existence of multiple approximately optimal weight vectors. MLLE addresses the regularization problem of standard LLE by introducing multiple weight vectors in each neighborhood rather than a single one, which makes it more stable and robust. MLLE minimizes the following embedding

cost function:

$$min \sum_{i=1}^{N} \sum_{l=1}^{s_i} \left\| t_i - \sum_{j \in J_i} w_{ji}^{(l)} t_j \right\|^2, s.t. TT^T = I \qquad (3)$$

$w_i^{(1)}, \ldots, w_i^{s_i}$ $(s_i \leq k_i)$ are linearly independent weight vectors, which are obtained from a matrix of the difference between data point $x_i$ and its $k_i$ neighbors.

In STREAM, the neighborhood size is chosen based on the number of cells and is set by default to 10% of the total number of cells. The number of MLLE components depends on the number of branches and on the complexity of the structure to learn. Typically, three components capture the main structure for most datasets (and this space can be readily visualized); however, increasing them may recover finer structures as discussed in main text under the section "STREAM trajectory inference in high dimensional spaces".

**Seeding initial tree structure.** The principal graph inference is based on a greedy optimization procedure that may lead to local minima, therefore in STREAM we proposed an initialization procedure that improves the quality of the inferred solutions and that speed up convergence. First, cells are clustered in the MLLE space using the affinity propagation method[53]. Affinity propagation is based on the idea of message-passing between sample points and finds a small set of exemplars, which are considered to be most representative of the other samples. In STREAM we use the scikit-learn implementation[54] with a damping factor set to 0.75. Based on the exemplars obtained, a minimum spanning tree (MST) is constructed using the Kruskal's algorithm. The obtained tree is then used as initial tree structure for the ElPiGraph procedure.

To learn principal graph in high-dimensional spaces, the procedure is modified as follows. Let $H$ be the number of MLLE components. We first apply the same strategy described above using the top $L$ components (~2–5) to obtain a tree in a low dimensional subspace. Based on this tree we calculate a principal graph in the $L$-dimensional space. Next, we map the learnt principal graph to the $H$-dimensional space as follows: for the coordinates of each node in the $H$-dimensional space, we obtain the first $L$ coordinates from the $L$-dimensional space. For the other $H-L$ coordinates, we use the mean values of the coordinate of the cells assigned to a given node. If a given node does not have any assigned cells, we infer the coordinates using k-nearest-neighbor strategy, i.e., the mean value of the nearest k cells is used to infer the node coordinate. The edges are instead preserved from the $L$-dimensional space. The node coordinate and edge information are then used to initialize a second-round of elastic principal graph inference in the $H$-dimensional space (Supplementary Fig. 4a).

**Elastic principal graph.** Elastic principal graphs are structured data approximators[27–29], consisting of vertices connected by edges. The vertices are embedded into the space of the data, minimizing the mean squared distance (MSD) to the data points, similarly to k-means. Unlike unstructured k-means, the edges connecting the vertices are used to define an elastic energy term. The elastic energy term and MSD are used to create penalties for edge stretching and bending of branches. To find the optimal graph structure, ElPiGraph uses a topological grammar (or, graph grammar) approach, which is described below. The core algorithm behind ElPiGraph was introduced and tested in publications preceding the development of STREAM[27–29]. However, the algorithm was further extended with domain-specific functions for single-cell data (described in the section Domain-specific optimization introduced to model single-cell). These functions are used by STREAM to improve the accuracy of the pseudotime and of the inferred trajectories.

Briefly, an elastic principal graph is an undirected graph with a set of vertices $V$ and a set of edges $E$. The set of vertices $V$ is embedded in the multidimensional space by minimizing the sum of the data approximation term and the graph elastic energy defined as:

$$U^{\phi}(X, G) = \frac{1}{|X|} \sum_{j=1}^{|V|} \sum_{i: P(i)=j} min\left( \left\| X_i - \phi(V_j)^2, R_0^2 \right\| \right)$$
$$+ \sum_{E^{(i)}} \left[ \lambda + \alpha \left( max\left(2, deg(E^{(i)}(0)), deg(E^{(i)}(1))\right) - 2 \right) \right] \left( \phi(E^{(i)}(0)) - \phi(E^{(i)}(1)) \right)^2$$
$$+ \mu \sum_{S^{(j)}} \left( \phi(S^{(j)}(0)) - \frac{1}{deg(S^{(j)}(0))} \sum_{i=1}^{deg(S^{(j)}(0))} \phi(S^{(j)}(i)) \right)^2$$
$$(4)$$

where $X = \{X_i\}, i = 1 \ldots |X|$ is the set of data points, $E^{(i)}(0)$ and $E^{(i)}(1)$ denote the two vertices of a graph edge $E^{(i)}$, and $S^{(j)}(0), \ldots, S^{(j)}(k)$ denote the vertices of a star $S^{(j)}$ in the graph (where $S^{(j)}(0)$ is the central vertex, to which all other vertices are connected), $deg(V_i)$ is a function returning the order $k$ of the star with the central vertex $V_i$, $\varphi(V_j)$ is the mapping function $\varphi: V \rightarrow R^m$, which defines a position of the $j^{th}$ graph vertex in the multidimensional space of data, $P(i) = arg \min_{j=1 \ldots |V|} X_i - \phi(V_j)$ is a data point partitioning function associating each data point $X_i$ to the closest vertex index. Finally, $R_0$, $\lambda$, $\mu$, and $\alpha$ are parameters having the following meaning: $R_0$ is the trimming radius such that points further than $R_0$ from any node do not contribute to the optimization of the

graph, $\lambda$ is the edge stretching elasticity modulo regularizing the total length of the graph edges and making their distribution close to equidistant in the multidimensional space, $\mu$ is the star bending elasticity modulo controlling the deviation of the graph stars from harmonic configurations. For any star $S^{(j)}$, if the embedding of its central vertex coincides with the mean of its leaf embedding, the configuration is considered harmonic. $\alpha$ is a coefficient, which allows controlling the topological complexity of the resulting graph and is helpful in controlling the branching potential.

Given a set of data points and a principal graph with nodes embedded into the original data space, a local minimum of $U^{\varphi}(X,G)$ can found by applying a splitting-type algorithm. Briefly, at each iteration given the initial guess of $\varphi$, the partitioning $P(i)$ is computed, and then, given the $P(i)$ partitioning, $U^{\varphi}(X,G)$ is minimized by finding new node positions in the data space. A remarkable feature of ElPigraph is that the $U^{\varphi}(X,G)$ minimization problem is quadratic with respect to vertices coordinates and can be solved in a very efficient way by computing the roots of a system of linear equations. Importantly, the convergence of this algorithm has been proven[29,55].

The most innovative aspect of ElPiGraph, when compared to other algorithms, is the use of topological grammars to more extensively explore the space of possible graph structures. Briefly, topological grammar rules define a set of possible transformations of the current graph topology. Afterwards, the graph configuration of this set possessing the minimal energy $U^{\varphi}(X,G)$ after fitting the candidate graph structures to the data is chosen as the locally best with a given number of nodes. Topological grammars are then iteratively applied to the selected graph until given conditions are met (e.g., a fixed number of grammar application, or a given number of nodes is reached). The graph learning process is in principle similar to a gradient-based descent approach in the space of all possible graph structures achievable by applying a set of topological grammar rules (e.g., in the set of all possible trees). Finally, the use of ElPiGraph results in an explicit definition of the principal tree embedded into the data space. The explicit tree structure can be studied independently on the data, or the data can be mapped onto the principal tree and studied in its intrinsic, geodesic, coordinates. A detailed description of ElPiGraph and related elastic principal graph approaches is available elsewhere[27–29].

Concerning the choice of the parameter values, in STREAM these default values are used for the principal graph inference with ElPiGraph: $R_0 = \infty$, $\alpha = 0.02$, $\mu = 0.1$, $\lambda = 0.02$. These values worked well across all the single-cell datasets tested. However, if necessary, these parameters can be easily modified by the user directly from the STREAM package. The ElPiGraph.R R package, used as part of STREAM to fit principal trees to the data, is available at https://github.com/sysbio-curie/ElPiGraph.R.

**Domain-specific optimizations to model single-cell data.** Although ElPiGraph is a general approach to construct principal trees (and other topologies), the obtained structures may not optimally describe biologically relevant trajectories or accurately capture pseudotime information based on single-cell data. Therefore, in addition to the described seeding strategy, in STREAM several single-cell specific optimizations were introduced to the core algorithm of ElPiGraph (Supplementary Fig. 1):

*Control over-branching*: A regularization parameter $\alpha$ with range $(0,1]$ was introduced to explicitly control the complexity of the resulting graph structure. Larger values of $\alpha$ lower the propensity of ElPiGraph to introduce branching points. An extreme value close to one prevents the creation of new branches not present in the initial seeding structure. Users can control this parameter based on the expected characteristics of noise and dimensionality of the data. By default, $\alpha$ is set 0.02 and this value was used for all the analyses performed.

*Prune branches*: The standard elastic principal graph favors harmonicity, i.e., star-shape subgraphs with a central node connected with equally spaced nodes. We have observed that this may lead to trivial branches with few cells. With the pruning grammar rule, STREAM is able to remove branches that are either associated with an excessively small number of cells or that are shorter than a minimal length. This step helps to get rid of spurious and unnecessary branching events that may not reflect real developmental trajectories.

*Shift branching nodes*: To minimize the elastic principal graph energy, ElPiGraph balances the reconstruction error, total length of edges, and the graph harmonicity. However, it may happen that the optimal solution places branching nodes to low-density regions with few cells. This can be hard to interpret biologically since these nodes should correspond to branching cell states within the cell population. With the shift branching grammar rule, each branching node is repositioned to the closest area with higher cell density to better match the most plausible region corresponding to the true branching event.

*Finetune branching nodes*: The standard elastic principal graph procedure well summarizes the principal structure. However, branching sections (i.e., regions close to branching events) may be not described in sufficient details by the obtained curves due to the limited number of nodes used around branching nodes by the global optimization. This grammar rule is able to optimize the space around branching nodes by locally adding a set of nodes in their proximity to better characterize branching events and to improve the pseudotime inference.

*Extend leaf nodes*: The standard elastic principal graph penalizes the total edge length to more robustly capture the main underlying structure. Although this

strategy works well when optimizing the graph structure, it may lead to border effects when projecting the data onto the leaf nodes. In fact, edges connecting internal node to leaf nodes rarely extend to the border of their local cloud of points. This is not ideal in the context of pseudotime reconstruction since multiple cells would be mapped to the same leaf node and assigned the same pseudotime. Hence, we extend the principal tree by attaching a new node to each leaf. The location of each node is based on the distribution of points around the corresponding leaf node. This enables the principal graph to better cover terminal cells and infer more accurately their pseudotime.

These graph grammar rules correspond to separate functions in STREAM and their usage is described in Supplementary Note 5.

**Flat tree plot**. The tree structure learned in the 3D space (or higher dimensional space), is first approximated by linear segments (each representing a branch) and mapped to a 2D plane based on a modified version of the force-directed layout Fruchterman-Reingold algorithm[56]. We adjust each edge length in order to preserve the lengths of the branches of the original tree. Finally, using both the pseudotime location on the assigned branch and the distance from it in the MLLE space, we map cells to the obtained tree in the 2D plane. Cells are represented as dots and randomly placed to either side of the assigned branches. Each node in the tree indicates one cell state (cell states are sequentially named S0, S1, … starting from a randomly selected node) and the resulting structure is called flat tree plot.

**Subway map plot**. Starting from the flat tree plot and with a designated root or start node, breadth-first search is used to order and arrange nodes and edges horizontally on a 2D plane. Because we preserve the branch lengths of the original tree, the x-axis represents the distance (namely pseudotime) from the start node along the different branches. Cells are then mapped to the obtained structure, called subway map plot with the same strategy used for the flat tree plot. To display gene expression, each cell is colored according to its gene expression (the maximum value in the colormap is set as 90 percentile of gene expression values across all cells).

**Stream plot**. Starting from the subway map plot, for each cell type (if cell labels are provided), using a sliding window approach, we first calculate the number of cells in each window along a developmental branch. To provide smooth transitions around the branching nodes, in those regions the sliding window spans both parent branch and children branches and then proceeds independently on each branch. Then, the numbers of cells in all sliding windows are normalized based on the length of the longest path in the tree. The vertical layout of different branches is optimized by taking into consideration normalized numbers of cells to make sure there will not be overlap between branches. Based on the normalized sliding window values, we first use linear interpolation to construct a set of supporting points. Then the Savitzky-Golay filter (a smoothing filter able to preserve well the signal and avoid oscillations)[57] is applied to create smooth curves based on the set of supporting points. Finally, the obtained curves polygons (one for each cell type) are assembled to form the stream plot. On the stream plot, the length of each branch is the same as in the subway map plot and represents pseudotime, whereas the width is proportional to the number of cells at a given position. To display gene expression, we consider, for each sliding window, not only the number of cells but also their average gene expression values smoothed by bicubic interpolation (the maximum value is set as the nintieth percentile of the average gene expression values from all the sliding windows).

**Diverging gene detection**. For each pair of branches $B_i$ and $B_j$, and for the gene $E$, the gene expression values across cells from both branches are scaled to the range [0,1]. For gene expression $E_i$ from $B_i$ and gene expression $E_j$ from $B_j$, we first calculate their mean values. Then, we check the fold change between mean values to make sure it is above a specified threshold (the default log2 fold change value is >0.25). Mann–Whitney U test is then used to test whether $E_i$ is greater than $E_j$ or $E_i$ is less than $E_j$. Since the statistic $U$ could be approximated by a normal distribution for large samples, and $U$ depends on specific datasets, we standardize $U$ to Z-score to make it comparable between different datasets. For small samples where this test is underpowered (<20 cells per branch), we report only the fold change to qualitatively evaluate the difference between $E_i$ and $E_j$. Genes with Z-score or fold change greater than the specified threshold (2.0 by default) are considered as differentially expressed genes between two branches. Formally,

$$z = \frac{U - m_U}{\sigma_U} \tag{5}$$

Where $m_U$, $\sigma_U$ are the mean and standard deviation, and

$$m_U = \frac{n_i n_j}{2} \tag{6}$$

$$\sigma_U = \sqrt{\frac{n_i n_j}{12}\left((n+1) - \sum_{l=1}^{k}\frac{t_l^3 - t_l}{n(n-1)}\right)} \tag{7}$$

Where $n = n_i + n_j$ $n_i, n_j$ are the number of cells in each branch, $t_i$ is the number of cells sharing rank $l$ and $k$ is the number of distinct ranks.

**Transition gene detection**. For each branch $B_i$ and for each gene $E$ we first scale the gene expression values to [0,1] for convenience. Then we check if the candidate gene has a reasonable dynamic range considering cells close to the start and end points. To this end, we consider the fold change in average gene expressions of the first 20% and the last 80% of the cells based on the inferred pseudotime. If the difference is greater than a specified threshold (the default log2 fold change value is 0.25), we then calculate Spearman's rank correlation between inferred pseudotime and gene expression of all the cells along $B_i$. Genes with Spearman's correlation coefficient above a specified threshold (0.4 by default) are identified and reported as transition genes.

**Leaf gene detection**. For each gene $E$ we scale the gene expression values to [0,1]. Then we calculate the average gene expressions for all leaf branches. Based on the average expressions, we calculate the Z-scores of all leaf branches. If there is any leaf branch with an absolute Z-score greater than 1.5, then the leaf branch with the highest absolute Z-score value will be picked as the candidate leaf branch. Next, Kruskal–Wallis H-test is computed for all the leaf branches to test if a significant difference of gene expression median value between leaf branches exists. If it is significant (p-value < 0.01), then a post-hoc pairwise Conover's test is computed for multiple comparisons of mean rank sums test between all leaf branches. If the p-values between the candidate leaf branch and the other leaf branches are all below the specified threshold (0.01), then the gene $E$ will be considered as leaf gene of the candidate leaf branch.

**Mapping procedure**. The mapping feature aims to map new cells to an inferred principal tree. For a set of unmapped cells X = $\{x_i | i = 1,…,M\}$ and a fitted tree $T$ built using the set of cells Y = $\{y_j | j = 1,…,N\}$, we assume that X and Y have the same measured genes and are sequenced using the same experimental protocol. We also assume that both X and Y are library size normalized, log2 transformed if necessary and that batch effects have been removed. To map cell $x_i$ into the embedding, we first find its nearest $K$ neighbors in Y, based on the same feature genes and $K$ used to build $T$. The largest distance between $x_i$ and its $K$ neighbors is then chosen as the radius $r$. Then all the cells in Y within the radius $J_i = \{y_j | d(x_i, y_j) \le r\}$ are used to compute a set of weights $W_i = \{w_{ji}, j \in J_i\}$ as defined in the original MLLE procedure. Finally, using the MLLE embedding vectors $V = \{v_1,…,v_N\}$, the new cell position $x'_i$ is calculated in the embedding with the following equation:

$$x'_i = \sum_{j \in J_i} v_j \times w_{ji} \tag{8}$$

After mapping, each cell is assigned to its closest branch in T.

Although this procedure is helpful to compare different conditions, there are some important points to consider. The mapping feature has some intrinsic limitations since it cannot introduce new fate branches in addition to the ones already present in the reference principal tree. In this case, pooling all the cells together and re-computing the trajectories may be needed to uncover the new fates. Thanks to the provided visualization tools, it is easy for the user to check if a new potential branch may be necessary to better describe the new cells. In fact, in both the flat tree and the subway map plots, the distances between cells and branches are inherited from the original space so it is easy to determine the confidence in assigning a given group of cells to their closest branch. If the new cells have much larger distances than the reference cells in any given branch, this will suggest that the built trajectory might not cover all the potential fates.

**STREAM analysis on scATAC-seq data**. For the scATAC-seq analysis, a total of 3,072 cells were profiled using FACS to isolate 9 distinct populations from CD34+ human bone marrow, which encompassed progenitors for four well-defined lineages[44]. Two thousand thirty-four high-quality cells passed quality control filtering and were used in the downstream analysis with STREAM. Specifically, cells were filtered so that 1000 unique nuclear fragments were observed for each cell and at least 60% of these reads aligned in open chromatin peaks. After filtering low quality cells, the mean intensity and GC content for each peak that was called for this dataset was computed using the addGCBias function for the hg19 genome using the BSgenome.Hsapiens.UCSC.hg19 package available through chromVAR[45]. These two coordinates were used to infer an empirically-defined set of background peaks to compute accessibility deviations, which have been described elsewhere[7,44]. As features we used an unbiased $k$-mer scoring, which is agnostic to any known transcription factor motif and thus generalizable to other systems. We used the matchKmers function in chromVAR with parameters k = 7 and genome = BSgenome.Hsapiens.UCSC.hg19, which returns a matrix of dimension number of peaks by number of $k$-mers where a 1 indicates that the peak contains the $k$-mer sequence. The output of this function was then included in the computeDeviations function to compute chromatin-accessibility z-scores for each of the $k$-mers in our dataset. This matrix of cells by $k$-mer accessibility z-scores serves as a data-driven dimensionality reduction of the chromatin-accessibility profiles of these cells. Based on the z-score matrix of $k$-mer DNA sequences, all the 7-mer features are standardized to have zero mean and unit variance. Since the z-score matrix of $k$-mers has both positive and negative values, the variable gene selection method based on gene expression

values is not directly applicable. As such, PCA is performed on the scaled matrix to convert z-score to principal components. According to the variance ratio elbow plot we selected the top 15 PCs but excluded the first component since it captured technical noise. Then the selected PCs are used as features for MLLE to reduce dimensionality. In the reduced MLLE space, the same strategy is used to reconstruct trajectories as previously described. Diverging and transition *k*-mers were selected with the same procedures used for gene selection. Finally, detected *k*-mers were mapped to known transcription factors using Tomtom[58](http://meme-suite.org/tools/tomtom) and a motif database previously assembled (github.com/buenrostrolab/chromVARmotifs)[44].

**Comparison on simulated datasets**. Given a set of *n* cells and assuming we know their developmental/sampling time and topological organization, i.e., how they are organized in branches, we can easily evaluate a generic reconstruction method with the following two metrics:

Difference between the number of inferred and true branches.

Correlation between the true sampling time *X* and the inferred pseudotime *Y*. For the pseudotime we use either the proposed ranking or the actual distance from the starting point as provided by each method. We used three different measure of correlation: Pearson correlation *r*, Spearman correlation *ρ*, and Kendall's tau correlation *τ*, calculated as follow:

$$r_{pseudotime} = \frac{\sum_{i=1}^{n}(x_i - \bar{x})(y_i - \bar{y})}{\sqrt{\sum_{i=1}^{n}(x_i - \bar{x})^2}\sqrt{\sum_{i=1}^{n}(y_i - \bar{y})^2}} \quad (9)$$

$$r_{rank} = \frac{\sum_{i=1}^{n}(x_i - \bar{x})\left(rg_{y_i} - \overline{rg_y}\right)}{\sqrt{\sum_{i=1}^{n}(x_i - \bar{x})^2}\sqrt{\sum_{i=1}^{n}\left(rg_{y_i} - \overline{rg_y}\right)^2}} \quad (10)$$

$$\rho = \frac{cov(rg_X, rg_Y)}{\sigma_{rgX}\sigma_{rgY}} \quad (11)$$

$$\tau = \frac{1}{n(n-1)}\sum_{i \neq j} sgn\left(x_i - x_j\right) sgn\left(y_i - y_j\right) \quad (12)$$

Where $rg_X$ and $rg_Y$ are the ranks of cells, $cov\,(rg_X, rg_Y)$ is the covariance of rank variables, $\sigma_{rgx}$ and $\sigma_{rgy}$ are the standard deviations of rank variables. Note that since both Spearman correlation *ρ* and Kendall's tau correlation *τ* are rank-based methods, the correlation between *X* and *Y* and the correlation between *X* and $rg_Y$ are the same, so we consider only the correlation between *X* and *Y*.

**Comparison on real datasets**. To evaluate the quality of reconstruction in real datasets in which we do not have the real developmental time and topological information, we used the following two metrics:

(1) *Path-specific marker gene correlation analysis*: In real datasets oftentimes, we do not have the sampling time along a branch. In this case, instead, it is helpful to evaluate how the inferred pseudotime recapitulates the progressive activation or repression of an important gene along that branch. The main idea here is that ordering cells based on a marker gene, which is important in defining a developmental trajectory, as a reasonable surrogate for the correct pseudotime ordering. As in the simulation case we computed four correlation coefficients using marker gene expression *X* and the inferred pseudotime *Y*.

(2) *F₁ score analysis on diverging or mutually exclusive marker genes*: Let us consider a pair of diverging or mutually exclusive marker genes, $G_i$ and $G_j$. These genes should be highly expressed on different committed branches and rarely co-expressed in the same cell. We define $B_i$ as the branch, which contains the most cells express $G_i$. Then we can define as true positive (TP) for $B_i$ the number of cells expressing $G_i$. The number of cells expressing $G_i$ on the other branches is defined as false negative (FN). The number of cells expressing $G_j$ on $B_i$ is defined as false positive (FP). Similarly, for $G_j$, $B_j$ is the branch, which has the most cells expressing $G_j$. TP is the number of cells expressing $G_j$ on $B_j$. FN is the number of cells expressing $G_j$ on the other branches. FP is the number of cells expressing $G_i$ on $B_j$. Based on the following equations, recall, precision and F1 score are calculated, respectively, for $G_i$ and $G_j$ as follow:

$$Recall = \frac{TP}{TP + FN} \quad (13)$$

$$Precision = \frac{TP}{TP + FP} \quad (14)$$

$$F_1 = 2 \times \frac{Precision \times recall}{Precision + recall} \quad (15)$$

## Data availability
The authors declare that the data supporting the findings of this study are available within the paper and its supplementary information files (Supplementary Data 1 and 2).

## Code availability
STREAM is available as a user-friendly open-source software and can be used interactively as a web-application at http://stream.pinellolab.org (Supplementary Fig. 11, Supplementary Note 4), a bioconda package 'stream' for step-by-step analysis https://bioconda.github.io/recipes/stream/README.html (Supplementary Note 5), or as a standalone command-line tool: https://github.com/pinellolab/STREAM (Supplementary Note 6). All the analyses presented in this manuscript can be reproduced using the bioconda package and the provided Jupyter notebooks in Supplementary Data 1 and 2.

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

## Acknowledgements

This project has been made possible in part by grant number 2018- 182734 to L.P. from the Chan Zuckerberg Initiative DAF, an advised fund of Silicon Valley Community Foundation. L.P. is also partially supported by a National Human Genome Research Institute (NHGRI) Career Development Award (R00HG008399). G.Y.'s research was supported by a Claudia Adams Barr Award and a Bridge Award. J.D.B. acknowledges support from the Harvard Society of Fellows and Broad Institute Fellowship. J.D.B. also acknowledges the Allen Distinguished Investigator Program, through The Paul G. Allen Frontiers Group for funding. A.Z. and L.A. were supported by ITMO Cancer SysBio program (MOSAIC) and INCa PLBIO program (CALYS, INCA_11692). D.M.L was supported by R24OD016761 and R01CA211734. D.E.B. was supported by NHLBI (DP2OD022716, P01HL032262) and the Burroughs Wellcome Fund. A.N.G was supported by Ministry of Education and Science of Russia (Project No. 14.Y26.31.0022). J.G. was supported by National Natural Science Foundation of China (NSFC) (grant No. 61772367). S.Z. was supported by the National Key Research and Development Program of China (grant No. 2016YFC0901704). We thank Stuart H Orkin, Luca Biasco, Danilo Pellin, Ruben Dries, Sara Garcia, Micheal Vinyard, and the members of the Pinello Lab for helpful discussions. We also thank P. G. Camara and R. Rabadan for sharing both simulation code and data. We also thank V. Svensson for helpful discussions regarding GPFates. We also thank F. Theis and L. Haghverdi for the suggestion on adapting DPT on sc-qPCR data. We also thank Xiaojie Qiu for sharing the data and the scripts to reproduce Monocle2 analyses (Fig. S16, PMID: 28825705). We also thank Johannes Köster and the bioconda team for helping us in the development of the bioconda stream package. Schematic panels from Fig. 6a were modified from Buenrostro et al., 2018 Cell.

## Author contributions

L.P. and G.Y. conceived this project. L.P., H.C., J.Y.H., L.A. and A.Z. created STREAM. L.P. and H.C. wrote the manuscript with input from L.A., J.Y.H., C.A.L., G.B., J.G., S.Z., A.N.G., D.E.B., M.J.A., D.M.L., A.Z., J.D.B. and G.Y.

## Additional information

**Competing interests:** The authors declare no competing interests.

