## [Peer Review File · Nature Communications]

Reviewers' comments:

Reviewer #1 (Remarks to the Author):

In this paper, Chen et al propose a new method to reconstruct differentiation trajectories from single-cell transcriptomics and epigenomics data called STREAM. In addition to the trajectory reconstruction, the method provides new visualization tools and can be easily run through a website. The authors test their methods on previously published datasets as well as on a simulated data, comparing the performance of STREAM to 10 alternative methods for trajectory inference.

The reconstruction of single-cell trajectories is an important and still open problem in the field of single-cell genomics, and the STREAM method introduces important improvements with respect to previous methods in terms of performance as well as usability and visualization of results; moreover, its description is clear and well written.

Hence, I recommend publication of the paper in Nature Communications, provided that the following points are addressed:

1. The paper starts with a general description of the features of STREAM. This sounds more like a conclusive rather than an introductory paragraph. To make the paper accessible to a broader audience, it would be better to start with a general introduction about single-cell trajectory inference (what it means, why it's important, what is missing from currently available methods).
2. The stream plots showing gene expression levels should indicate in the legend what the color gradient represents (I guess it's log₂ normalized expression?)
3. The comparison with previous methods should include more information about how the other methods were run: i.e., was the same input used for all of them (log₂ normalized expression)? Was this the normalization recommended in the original papers? Did the authors choose default values for all the parameters? Did they use the same distance metric (Euclidean)? ... A more thorough description for each of the alternative method is required, to ensure reproducibility and to show that the benchmark was run fairly across such different methods. In addition to this, the authors should provide the code they used to run all the methods on the different datasets.
4. Some results on the comparison with the previously published methods should be reported in the main figures, as this is an important point to show.
5. In Supp. Figure 7, what criteria were used to assess the "user-friendliness" of the different methods?

Reviewer #2 (Remarks to the Author):

Single-cell sequencing technologies enable the reconstruction of developmental trajectories, but trajectory reconstruction from epigenomic data and trajectory visualization are still challenging. In this paper, authors developed a new pipeline and interactive web named STREAM to construct single cell trajectories using a newly developed algorithm EIPiGraph. It is the first pipeline tailored for single cell ATAC-seq data. They provided a mapping procedure, mapping new cells to pre-computed structure without re-computing trajectories, which facilitated the study of perturbation effects and provided a way to validate the trajectories from unlabeled cells. They extended STREAM to epigenomics and firstly reconstructed trajectories from sc-ATAC seq data. Finally, they systematically compare STREAM with ten commonly used methods and concluded that STREAM consistently outperformed the others in all the comparisons. In general, it is potentially useful in many ongoing scRNA-seq studies, particularly in trajectory analysis.

Major comments:

1. A major concern, to my understanding, is that the core algorithm in STREAM for trajectory analysis is based on the paper Rizvi et al. "EIPiGraph: Robust And Scalable Learning Of Complex Dataset Topologies Via Elpigraph", which is in arxiv as a separate machine learning paper. Without

a peer-viewed paper, it is not convincing EIPiGraph is working as claimed. In addition, the novelty of this paper is much weakened if the core component of STREAM EIPiGraph is presented elsewhere as a separate machine learning paper. The method description and the comparison results with existing trajectory analysis are a bit misleading and unnecessary if EpiGraph is not firstly presented in this paper. Authors need to clarify the overlapping between STREAM and EIPiGraph and the unique novelty of the current paper. The paper would be much strengthened if they combine the two papers (EIPiGraph and STREAM) into one.

2. The authors may overstate the advantages of STREAM in the comparison with other existing methods. How they applied each method on sc-seq data was not clear. Did they use the corresponding default settings? Or did they tune each method to optimize the results? Monocle 2 perform much better on HSMM data compared with the synthetic data by Rizvi et al. This indicated that methods perform differently on different dataset. It is too arbitrary to conclude that STREAM outperformed all the others. In addition, it would be good to test STREAM in resolving a complex hierarchy as Monocle2 did (PMID: 28825705, in figure S16). The paper mentioned: "Monocle2 tends to generate over-branched structures", which might be beneficial in analyzing complex-hierarchy data.

3. The title emphasizes the usage on omics data, but author only provided a very special pipeline for chromatin-accessibility data. It might be worthwhile to talk about its generalization to other omics data type, otherwise, the title might be misleading.

4. In the main test, authors mentioned that scATAC-seq human data are very challenging to analyze due to its ultra-high dimension and sparsity, but how exactly this new pipeline resolve the problem is not clearly stated. Did the filtering or using k-mer as features resolve the issue of dimension and sparsity entirely? Or the EIPiGraph is more robust for this kind of data than other algorithms? Why PCA is used as the 2nd step of dimension reduction for scATAC-seq data, but MLLS is used for transcriptomic data?

5. Mapping feature is a novel and useful function. But the authors should not overstate its roles. Because it excludes the probability that the new cells generated by a perturbation undergo totally a new fate.

6. Long description of EIPiGraph in online methods seems fairly similar to the contents in their previous paper Albergante et al. 2018, which they cited already. It's a little redundant. Authors provided a nice table to compare STREAM with ten other methods, but there is no comment about the comparisons of the algorithms to infer the trajectories. Some discussion comparing EIPiGraph with other algorithms may be helpful.

7. In the formula of elastic energy, is k pre-determined in the second term? In other words, do you only sum up the squared deviations from harmonicity for star with order k , or you sum up all stars with any k ? I presume you mean the latter, but either way, it's better to be clear.

8. Since the edges length will be interpreted as the similarity of two states, is it still fair to compare them if different shrinkage weight λ is applied?

9. When matching new cells, will the algorithm force the new cell to be matched to at least one branch, even if new cells are quite different from training cells. Will batch effect affect matching? For example, new cells are from another individual with different sequencing depth or from a different scRNA technology. Is a normalization step needed before mapping?

10. In figure S6b, a group of Macrophages/Myeloid cells (grey) were wrongly reconstructed at the end of lymphoid lineage (S5). In lymphoid trajectory, T & B cells cannot be further separated. Authors may add a bit discussion to explain.

Minor comments:

1. Is sub title required in main text?

2. The notation k is used too often. It's first used to denote the other of star, and then is used to denote the data partition (capitalized though), and latter it also denotes the iterations.

Response to Reviewers

We thank the reviewers for carefully evaluating our manuscript and for the great suggestions that improved significantly the manuscript readability and the robustness of the STREAM software.

We believe that we have addressed all of their comments with our responses below, with substantial edits to the manuscript text, method section and supplement, and with the addition of new Supplementary Data 1-2, Supplementary Figures 4-6 and Supplementary Notes 4.

Importantly, to generate step-by-step reproducible results with STREAM we have also developed a new bioconda package along with 11 Jupyter Notebooks for all the analyses performed by STREAM. To make the comparison analysis more transparent we have also generated 24 Jupyter Notebooks for eight methods on three datasets to illustrate all the steps and parameters used (SCUBA is MATLAB-based so we provided a MATLAB script. TSCAN is website based and the analyses can be easily reproduced following the website tutorial).

We believe that this revised manuscript is greatly strengthened by these changes and appreciate the Reviewers' suggestions for improvement.

Reviewer #1 (Remarks to the Author):

In this paper, Chen et al propose a new method to reconstruct differentiation trajectories from single-cell transcriptomics and epigenomics data called STREAM. In addition to the trajectory reconstruction, the method provides new visualization tools and can be easily run through a website. The authors test their methods on previously published datasets as well as on a simulated data, comparing the performance of STREAM to 10 alternative methods for trajectory inference.

The reconstruction of single-cell trajectories is an important and still open problem in the field of single-cell genomics, and the STREAM method introduces important improvements with respect to previous methods in terms of performance as well as usability and visualization of results; moreover, its description is clear and well written.

Hence, I recommend publication of the paper in Nature Communications, provided that the following points are addressed:

- 1. The paper starts with a general description of the features of STREAM. This sounds more like a conclusive rather than an introductory paragraph. To make the paper accessible to a broader audience, it would be better to start with a general introduction about single-cell trajectory inference (what it means, why it's important, what is missing from currently available methods).*

We thank the reviewer for the suggestion. We have extended the manuscript by adding an introduction section and moved some of the more conclusive statements into the discussion section.

2. *The stream plots showing gene expression levels should indicate in the legend what the color gradient represents (I guess it's log2 normalized expression?)*

This is a great point. The color gradient may represent different units depending on the dataset analyzed. Therefore, we have revised the legends by adding the text 'gene expression (log2 normalized)' or 'UMI' for the scRNA-seq analysis and 'scaled k-mer z-score' for the scATAC-seq analysis.

3. *The comparison with previous methods should include more information about how the other methods were run: i.e., was the same input used for all of them (log2 normalized expression)? Was this the normalization recommended in the original papers? Did the authors choose default values for all the parameters? Did they use the same distance metric (Euclidean?)? ... A more thorough description for each of the alternative method is required, to ensure reproducibility and to show that the benchmark was run fairly across such different methods. In addition to this, the authors should provide the code they used to run all the methods on the different datasets.*

We thank the reviewer for raising this important point. We also believe this is an important part of the manuscript therefore we have significantly expanded the manuscript with:

- A new extended section in the main text in which we describe each method and its limitation: *Comparison with other methods on synthetic and real datasets*
- A supplementary note (Supplementary Note 2) to describe in detail how the methods were used (input, parameters used, normalization, etc.)
- Importantly we now provide Jupyter Notebooks that documents and make our comparison analysis fully reproducible (Supplementary Data 2).

4. *Some results on the comparison with the previously published methods should be reported in the main figures, as this is an important point to show.*

We have moved as suggested some of the supplementary figures in the main text (now Figures 4 and 5)

5. *In Supp. Figure 7, what criteria were used to assess the "user-friendliness" of the different methods?*

We agree with the reviewer that the term "user-friendliness" was ambiguous. In the revised manuscript we modified the Supplementary Table 1 and replaced this term with *Ease of installation*. We also provide now in Supplementary Note 2 a description of each term used in this table.

Reviewer #2 (Remarks to the Author):

Single-cell sequencing technologies enable the reconstruction of developmental trajectories, but trajectory reconstruction from epigenomic data and trajectory visualization are still challenging. In this paper, authors developed a new pipeline and interactive web named STREAM to construct single cell trajectories using a newly developed algorithm EIPiGraph. It is the first pipeline tailored for single cell ATAC-seq data. They provided a mapping procedure, mapping new cells to pre-computed structure without re-computing trajectories, which facilitated the study of perturbation effects and provided a way to validate the trajectories from unlabeled cells. They extended STREAM to epigenomics and firstly reconstructed trajectories from scATAC-seq data. Finally, they systematically compare STREAM with ten commonly used methods and concluded that STREAM **consistently outperformed** the others in all the comparisons. In general, it is potentially useful in many ongoing scRNA-seq studies, particularly in trajectory analysis.

Major comments:

1. A major concern, to my understanding, is that the core algorithm in STREAM for trajectory analysis is based on the paper Rizvi et al. "EIPiGraph: Robust And Scalable Learning Of Complex Dataset Topologies Via Elpigraph", which is in arxiv as a separate machine learning paper. Without a peer-viewed paper, it is not convincing EIPiGraph is working as claimed. In addition, the novelty of this paper is much weakened if the core component of STREAM EIPiGraph is presented elsewhere as a separate machine learning paper. The method description and the comparison results with existing trajectory analysis are a bit misleading and unnecessary if EpiGraph is not firstly presented in this paper. Authors need to clarify the overlapping between STREAM and EIPigraph and the unique novelty of the current paper. The paper would be much strengthened if they combine the two papers (EIPiGraph and STREAM) into one.

We would like to thank the reviewer for raising this important point. Although we referenced for the first time EIPiGraph in this manuscript and in the preprint mentioned by the reviewer (*EIPiGraph: Robust And Scalable Learning Of Complex Dataset Topologies Via EIPiGraph*) the core approach of this general method for principal graph reconstruction (combination of graph grammars and elastic energy minimization) has already been presented and validated before, in peer-reviewed publications. In fact, the core algorithm of EIPiGraph has been published here:

1) A.N. Gorban, N.R. Sumner, and A.Y. Zinovyev, Topological grammars for data approximation, Applied Mathematics Letters, 20 (4) (2007), 382-386.

2) A.N. Gorban, A. Zinovyev, Principal manifolds and graphs in practice: from molecular biology to dynamical systems, International Journal of Neural Systems, Vol. 20, No. 3 (2010) 219–232.

3) A.N.Gorban, A.Zinovyev. Principal graphs and manifolds. In Handbook of Research on Machine Learning Applications and Trends: Algorithms, Methods, and Techniques, Emilio Soria Olivas et al. (eds). 2009. IGI Global, Hershey, PA, USA

4) A.N. Gorban, E.M. Mirkes, A Zinovyev. Robust principal graphs for data approximation(2017). Archives of Data Science, Series A 2 (1), 1-16.

Therefore, in the revised manuscript we focused on presenting and validating the problem specific developments of the EIPiGraph core algorithm presenting new features specifically designed for STREAM to optimally handle single cell data. These domain specific developments are validated in the current manuscript, as a part of the STREAM pipeline, by comparing with other tools having the objective to reconstruct branching pseudo-time from single cell data.

We have therefore restructured the method section referring to published studies when necessary to remove the overlap and introduced a new section: *Domain specific optimization introduced to model single-cell data* and a new supplementary figure (Supplementary Figure 1). In this section we explain five domain specific optimizations that can be used to better describe biologically relevant trajectories and more accurately capture pseudotime information based on single cell data.

In addition, regarding the overlap between STREAM and EIPiGraph, we would like to mention that although building an elastic principal graph is an important step of STREAM, we provide a complete end-to-end solution for trajectory inference, marker gene discovery and visualization. Building a solid pipeline required other key steps in addition to construction of the principal graph including: **feature selection** (most variable genes or top principal components), **dimensionality reduction** (the application of the modified version of the LLE is presented here for the first time), **seeding the initial structure** for principal graph reconstruction (a heuristic message-passing-based method for initial nodes and MST method for initial edges that significantly improve the inferred structure). In this revised manuscript this seeding strategy was further extended to **learn trajectories in high dimensional spaces** (see response 2. to Reviewer #2 and its application described in a new section in the main text called *STREAM trajectory inference in high dimensional spaces*). Importantly, we also introduced a **mapping feature** that is not available in other trajectory inference methods). In addition, we propose **several visualizations plots** and also an interactive website. None of these steps and features is a part of the EIPiGraph package, which is instead focused on a general-purpose and computationally efficient solution to build principal graphs.

Notably, the EIPiGraph preprint cited in the manuscript focuses on a set of innovative features present in the package that are currently not used by STREAM, such as building ensembles of principal trees and constructing from them a consensus principal graph. EIPiGraph is presented there as a general-purpose machine learning method which can be applied to data originating from different technological fields. Therefore, considering that the new revisions removed the previous perceived overlap and that the two manuscripts have distinct focus and target audiences, we believe that they clearly stand now as independent works.

2. The authors may overstate the advantages of STREAM in the comparison with other existing methods. How they applied each method on sc-seq data was not clear. Did they use the corresponding default settings? Or did they tune each method to optimize the results?

We thank the reviewer for raising this important point. Please refer to response 3. to Reviewer #1.

Monocle 2 perform much better on HSMM data compared with the synthetic data by Rizvi et al. This indicated that methods perform differently on different dataset. It is too arbitrary to conclude that STREAM outperformed all the others.

We agree that the statement “STREAM consistently outperforms the other methods” is an over-simplification, and that different methods could work better on different datasets. Therefore, we have replaced this statement with more precise and detailed conclusions.

In order to reach a fair and representative assessment, we compared the performance of the 10 different methods on a combination of synthetic and real datasets covering different technologies and using multiple metrics. For the synthetic data we wrote in the new version of the manuscript: “STREAM has the best performance for two out of four metrics (and importantly when using distance-based pseudotime) and second-best performance for the other two rank-based metrics (following scTDA in which this synthetic dataset was proposed) (Fig. 4b).”

Then in the summary of that section: “In summary, STREAM recovers the correct topology, has the overall best pseudotime reconstruction performance when distance-based pseudotime is used...”

For the scRNA-seq data using the myoblast differentiation system to score the methods given that three marker genes are evaluated across 4 metrics we used as an overall metric their average rank and wrote in the manuscript: “When combining all three marker genes, STREAM has the overall best performance (calculated as the average rank for the four proposed metrics)” The average rank for each method is now also shown on Figure 5d and the values for all the metrics for the individual genes are shown in Supplementary Figure 8.

For the qPCR dataset comparison using the mouse hematopoietic system we scored the methods based on the F1 score and wrote: “STREAM has the highest F1-score for both Gata1 and Pax5 among all the methods tested and balance well precision and recall (Fig. 5f)”

In the discussion to summarize those results we wrote: “Our comparison of STREAM with 10 other methods shows that it is among the top scoring methods on both synthetic and real datasets (best average ranking across several metrics and most balanced method for branching detection).”

We believe that the updated text more precisely reflects the performance of STREAM relatively to the other methods for the different datasets and metrics used.

In addition, it would be good to test STREAM in resolving a complex hierarchy as Monocle2 did (PMID: 28825705, in figure S16). The paper mentioned: “Monocle2 tends to generate over-branched structures”, which might be beneficial in analyzing complex-hierarchy data.

We thank the reviewer for this suggestion. We have performed the suggested analysis and compared it with the results obtained by Monocle 2 (PMID: 28825705, in figure S16). This challenging analysis pushed us to test and update STREAM with a novel seeding strategy (presented in the method section) in order to robustly perform trajectory inference in high dimensional spaces. With the default parameters and 10 components as in Monocle2, STREAM is able to reconstruct complex-hierarchy data with six distinct outcomes. The results are presented in the main text at the end of the new section called *STREAM trajectory inference in high dimensional spaces* and in Supplementary Figure 6. In the same section we also discuss how this strategy was also helpful in the reanalysis of the in-drop zebrafish dataset increasing the number of the dimensions (20) to better capture all the expected lineages (see response 10 to Reviewer #2). We also used this dataset to test and compare how Monocle 2 recovers trajectories in this case. With this dataset, Monocle2 fails to recover reasonable trajectories in both low and high dimensional spaces with clear inconsistencies as discussed in this section and as shown in Supplementary Figure 4.

3. The title emphasizes the usage on omics data, but author only provided a very special pipeline for chromatin-accessibility data. It might be worthwhile to talk about its generalization to other omics data type, otherwise, the title might be misleading.

As suggested by the reviewer, we expanded the discussion to explain how our methodology could be potentially adapted and extended to other omics data in addition to the two omics modalities already supported (transcriptomic and chromatin accessibility data).

4. In the main test, authors mentioned that scATAC-seq human data are very challenging to analyze due to its ultra-high dimension and sparsity, but how exactly this new pipeline resolve the problem is not clearly stated. Did the filtering or using k-mer as features resolve the issue of dimension and sparsity entirely? Or the EIPiGraph is more robust for this kind of data than other algorithms?

We thank the reviewer for the nice suggestion. Sparsity is intrinsic to single-cell ATAC-seq data since the signal is limited by the DNA copy number, which only consists of 0,1, or 2 reads within a diploid genome. In addition, compared to previously published datasets in other model organisms (mouse, *Drosophila melanogaster*), the human genome is considerably larger making this problem accentuated.

EIPiGraph is a method for principal graph estimation that is robust to noise but as mentioned before it is only a single step of the STREAM pipeline and it doesn't provide any direct solution to address the problems related to scATAC-seq data.

In STREAM we solve these problems before the principal graph inference step considering the following approach. To overcome the sparsity of the data and the limited ability to capture a given region in a single cell we focus on chromatin-accessibility variable regions across cells instead of scoring the entire genome, and then aggregate over general features

related to chromatin accessibility on these regions. To this end STREAM uses an unbiased set of DNA sequence features (i.e. k-mers) and calculate their accessibility deviations across cells. We believe this is a reasonable strategy to deal with the sparsity and the dimensionality problems. This is also an interpretable approach since it can be used to capture and map the important DNA sequence features that best explain the different trajectories to transcription factors. This idea is similar in spirit to a method proposed in a previous scRNA-seq study (Fan, J. et al. Characterizing transcriptional heterogeneity through pathway and gene set over dispersion analysis. *Nat Methods* 13, 241-244 (2016)). In this study they show that measuring dispersion over gene sets instead of individual genes is a powerful way to analyze sparse data. In addition, we have observed that selecting the top principal components of the scaled z-score matrix not only further reduces the dimensionality but also improves the recovered structures.

We have expanded the main text to summarize these key points in the manuscript.

Why PCA is used as the 2nd step of dimension reduction for scATAC-seq data, but MLLS is used for transcriptomic data?

We are sorry for the confusion, this has been clarified in the manuscript. MLLS is still used for scATAC-seq as a dimensionality reduction step. PCA is used as feature selection step before MLLS to select top principal components in place of the variable genes used for scRNA-seq data analysis. This is necessary since scATAC-seq datasets are modeled as a z-score matrices of k-mers, so the proposed gene selection procedure is not directly applicable in this context.

5. Mapping feature is a novel and useful function. But the authors should not overstate its roles. Because it excludes the probability that the new cells generated by a perturbation undergo totally a new fate.

We appreciate this insightful comment. It is true that a new fate may appear if the mapped cells are significantly different from the cells used to infer the reference structure. Our visualization tools (i.e. the Flat Tree and Subway Map plots) can be used to determine the confidence in assigning a cell to an existing branch (distances of cells from their closest branch). In fact, if cells should be assigned to a “missing” branch, these cells will have relatively large distances compared to the cells used to infer the trajectories. This information could be used as an indicator for new cell fates. In this case, the user may need to recompute the trajectories using all the cells. We added this sentence to the discussion:

“However, the mapping procedure may not be optimal if the new cells have totally different fates compared to the ones present in the reference structure as discussed in the Methods section.”

We also significantly expanded the mapping section in the method section to highlight this important point:

“Although this procedure is very helpful to compare different conditions, there are some important points to consider. The mapping feature has some intrinsic limitations since it

cannot introduce new fate branches in addition to the ones already present in the reference principal tree. In this case, pooling all the cells together and recomputing the trajectories may be needed to uncover the new fates. Thanks to the provided visualization tools it is easy for the user to check if a new potential branch may be necessary to better describe the new cells. In fact, in both the Flat Tree and the Subway Map plots, the distances between cells and branches are inherited from the original space so it is easy to determine the confidence in assigning a given group of cells to their closest branch. If the new cells have much larger distances than the reference cells in any given branch, this will suggest that the built trajectory might not cover all the potential fates.”

6. Long description of EIPiGraph in online methods seems fairly similar to the contents in their previous paper Albergante et al. 2018, which they cited already. It's a little redundant. Authors provided a nice table to compare STREAM with ten other methods, but there is no comment about the comparisons of the algorithms to infer the trajectories. Some discussion comparing EIPiGraph with other algorithms may be helpful.

We thank the reviewer for the nice suggestion. *EIPiGraph* as discussed before is a general method for principal graph inference. The method is not directly applicable to single-cell data but included here as one of 6 steps in our pipeline (see also response 1 to Reviewer #2). To avoid redundancy, we now present in the method section a more succinct description of *EIPiGraph* and reference the interested readers to other 4 studies in which the elastic principal graph methods are described in detail and compared with other general methods that may be used in the context of trajectory inference.

We have also added, as suggested, at the beginning of the new section *Comparison with other methods on synthetic and real datasets* a short discussion about the core algorithms used by each trajectory method and discussed their limitations compared to *EIPiGraph*.

“Monocle2 uses reversed graph embedding (by default DDRtree) to learn an explicit principal graph to describe the single-cell transcriptomic data. During the principal graph learning step, in each iteration, DDRtree moves cells to the nearest vertex, hence distorting the original configuration of cells in the manifold that may result in an uneven distribution with more cells close to vertices and fewer cells in between (Supplementary Figures 7,8). Mpath first clusters cells and designates landmarks. Based on the landmarks a weighted neighborhood network is constructed and subsequently trimmed to obtain a state transition network. Mpath requires prior information (e.g. FACS sorting and time points) to designate landmarks and the final transition networks are sensitive to the chosen landmarks. Wishbone builds a kNN graph based on the identified most relevant diffusion components by using gene set enrichment analysis (GSEA). Then a random sample of cells termed waypoints are used for ordering cells and branch identification based on the inconsistency between “waypoints”. Wishbone can only detect simple bifurcation (i.e. two cell fates). DPT first reduces the dimensionality using a diffusion map approach. Then in the diffusion map space a random-walk-based distance is computed. Branching points are identified by comparing two DPT orderings over cells. SLICER uses locally linear embedding (LLE) to reduce the dimension and then uses a KNN graph to order cells by the shortest path distance from root cell. Branches are detected based on geodesic entropy. All these three methods, i.e. Wishbone, DPT and SLICER cannot infer trajectories without specifying a start

cell. In addition, they cannot explicitly show cellular trajectories with a specific topological structure. Instead they simply visualize cells using the first two components of the dimensionality reduction method adopted. scTDA uses the Mapper algorithm to build a topological representation. Briefly, a low dimensional space (obtained for example by MDS) is first divided into overlapping bins and cells within each bin are clustered in the original space. The topological structure is obtained connecting clusters that share at least one cell. However, spurious edges may appear since an edge could be formed as long as two clusters have a non-empty intersection. In addition, the obtained topology may also depend on the binning strategy used. TSCAN applies principal component analysis to a gene-cluster level expression matrix. Then top PCs are selected to cluster cells and a MST is constructed based on the clustering solution. Finally, cells are projected and ordered on the tree structure. [By default, TSCAN does not report branching events, in fact reports only the linear path with the largest number of clusters. Also, since cells are projected to clusters and not along the edges, the pseudotime assignment is limited in resolution. SCUBA first constructs a binary tree by iteratively clustering and refines the tree based on a penalized likelihood function. Then it models gene dynamics using bifurcation theory. This method can infer multiple bifurcation events however cannot model branching events with more than two cells fates. GPfates uses OMGP (Overlapping Mixture of Gaussian Processes) to model the temporal dynamics. However, it requires a prespecified number of trends/trajectories. In addition, both SCUBA and GPfates requires temporal information (pseudotime) as input to model the dynamics. SCUBA uses principal curve in tSNE space to infer cell order. GPfates uses GPLVM (Gaussian Process Latent Variable Model) to infer pseudotime. PHATE is a visualization method that preserves well single-cell trajectory structure based on data diffusion but doesn't provide branch assignment and pseudotime information.]

7. In the formula of elastic energy, is k pre-determined in the second term? In other words, do you only sum up the squared deviations from harmonicity for star with order k , or you sum up all stars with any k ? I presume you mean the latter, but either way, it's better to be clear.

We thank the reviewer for pointing out this issue. Stars of all orders contribute to the definition of the elastic energy. In order to make it clear, we modified the formula and the text, using a sum over all stars in the graph and denoting their order by the $\text{deg}()$ function instead of k .

8. Since the edges length will be interpreted as the similarity of two states, is it still fair to compare them if different shrinkage weight λ is applied?

Reviewer refers to the notion of “state”, which we can understand in two ways. First, each graph node can be interpreted as a cell state. Below we argue that it is not necessarily the case for EIPiGraph-inferred trajectory and that one should avoid such interpretation. Second, a particular biological meaning can be assigned to only some nodes of the graph (i.e., terminal nodes). In this case we argue below that the estimated geodesic distance along reconstructed principal curve (the sum of all edges defining the curve) is not affected by the number of nodes/edges along it or the uniformity of their placement (controlled, in

particular, by lambda parameter). Changing lambda parameter won't have an effect of shrinking the total principal tree length, after application of the complete structure learning algorithm.

Some clarifying statements go below:

In EIPiGraph (unlike, for example, self-organizing maps), graph nodes do not necessarily represent the centers of local data clusters. By contrast, elastic principal graphs may even contain "empty" nodes not associated to any data point. In a sense, a branch of a principal tree represents a local principal curve as a one-dimensional centroid of the continuum of system states. We note that mapping of data is performed on the branches (edges) of the graph such that the data points can be mapped in between nodes, representing continuum of states.

The lambda term of the elastic energy favors principal graphs with nodes located at uniform distances one from each other, which renders data mapping onto the graph more representative of the multidimensional distance. This aspect has been described in previous publications (e.g., Gorban and Zinovyev. Elastic principal graphs and manifolds and their practical applications. Computing 75(4), 2005).

Such close-to-equidistant node distribution quantifies the pseudo-time (as an amount of transcriptomic changes along a geodesic path in the data point cloud) more objectively. Increasing lambda typically makes the nodes more densely packed along the principal tree branches and vice versa but does not necessarily changes the topology or smoothness of the principal tree (the latter is much more strongly controlled by the mu parameter). In this sense, nodes of the elastic principal graph form a close to equidistant grid landmarking branches of a cellular trajectory but their exact positions along the branches or their density do not have particular meaning themselves.

Certain nodes (e.g. terminal nodes associated with differentiated states or branching nodes), may have a particular biological meaning and can potentially be interpreted as particular states of the system. In this context, it is possible to compare distances for these nodes along the local principal curves that connect them even if the principal curves with different values of lambda may have a different number of nodes or edges.

We also want to highlight that in STREAM the lambda parameter was fixed to 0.02 for all the analyses performed.

9. When matching new cells, will the algorithm force the new cell to be matched to at least one branch, even if new cells are quite different from training cells. Will batch effect affect matching? For example, new cells are from another individual with different sequencing depth or from a different scRNA technology. Is a normalization step needed before mapping?

We thank the reviewer for raising these important points. When matching new cells, STREAM will map each cell to its closest branch in the reference structure. In addition, we can easily estimate the reliability of the mapping procedure based on the distance from new cells to the branches as discussed in response 5 to Reviewer #2.

Regarding normalization, batch effects and sequencing depth we assume that the appropriate procedures are applied before using the mapping feature. Also, we assume that the same scRNAseq technology is used to generate the data.

We expanded the Mapping procedure under Methods to highlight these important points.

10. In figure S6b, a group of Macrophages/Myeloid cells (grey) were wrongly reconstructed at the end of lymphoid lineage (S5). In lymphoid trajectory, T & B cells cannot be further separated. Authors may add a bit discussion to explain.

To better annotate the cell lineage trajectories in zebrafish marrow-derived blood cells, we have now refined our analysis by increasing the number of dimensions in accordance with our previous publication (Tang et al, JEM 2017). Specifically, we have used 20 dimensions to better reconstruct lineage trajectories with STREAM. These data are now shown in Supplemental Figure 4,5.

Using this refined analysis, we can now clearly separate T and B cell subsets as nicely suggested by the reviewer. Moreover, these analyses separate our previously annotated “Macrophage/myeloid” cell population away from T and B lymphocytes. This population of cells was previously described by Tang et al., JEM 2017 as a novel cell type for which the cell trajectories failed to place it into either the myeloid or lymphoid lineage. Our modified cell lineage tree now suggests that these rare cell types, comprising less than 1.36% of the marrow, may be derived from lymphoid precursor cells. The original study could not assign these unique cell types to arising in either the myeloid or lymphoid cell lineages. By contrast, we observed that most of those cells were more similar to cells in the lymphoid branch. These results were verified using independent analysis using UMAP and Monocle2 (Supplementary Fig. 4), each of which confirmed shared lineage with lymphoid cells.

These cells however express also macrophage marker genes including *mpeg1.1*, *irf8*, *ctsbb*, *ccr9b*, *tlr7*, *ccl39.3*, and *p2rx3b* that are not expressed in the main clade capturing the cells labeled as Macrophages. On the contrary, we observed genes in the main Macrophages clade that are not expressed in this rare subpopulation such as *mfp4*, *grn1* or *marco* (Supplementary Fig. 5). These cells also share remarkable similarity with other myeloid cells as previously defined by Tang et al., 2017. Taken together, we believe that further experimental analyses are necessary to fully characterize this rare and distinct population.

Our new analysis also further subdivided both the neutrophil and macrophage lineages. Diversity in these lineages is well-known in mouse and human. For example, N1 and N2 neutrophils exert a wide diversity of cellular responses in cancer following polarization into each cell state by TGF- β and type 1 interferons, respectively (Hong et al., 2017). Subsets of neutrophils also have either immune-stimulatory or anti-inflammatory function reflecting a wide function and cellular diversity of cell states (Wang et al., 2018). Further diversity of cell states includes characterizing inflammatory neutrophils as 1) immature, 2) resting, 3) primed, and 4) active (see figure 2 below, reproduced from Hong et al., 2017). Moreover, our analysis of macrophages identified three distinct subtypes of cells. Although controversial, it is well-appreciated that macrophages have a diversity of cellular states including M1 and M2 macrophage-states that exhibit anti and pro-inflammatory roles in cancer. These cell states are now appreciated to be highly plastic and heterogeneity can be seen based on the age of animal, infection/disease type, and tissue in which macrophages become active (Cochain et al., 2018; Bonnardel and Guilliams, 2018; and Kiss et al, 2018). Moreover, macrophages commonly adopt unique tissue-resident cell states, including those in the kidney (Munro and Hughes, 2017), suggesting that the marrow of zebrafish may comprise both classically-defined blood macrophage subsets and also tissue-resident macrophages with local regulatory roles in inflammatory responses that occur in the kidney. Indeed, macrophage subsets have been identified in zebrafish that have important functional differences (Lu et al, Journal of Immunology 2017 and Nguyen-Chi et al., eLife, 2015). Notably, several genes that are uniquely expressed in these sub-branches of the neutrophil and macrophage lineages are shown in Supplementary Figure 5.

In total, our results are exciting since these myeloid cell subsets are largely ill-defined in the zebrafish model. Careful cell lineage tracing, IHC, and functional experiments will be required to better characterize these subpopulations, which is well beyond the scope of the current manuscript.

Minor comments:

1. Is sub title required in main text?

We thank the reviewer for the nice suggestion. Yes, we have restructured the manuscript by adding sub titles to improve its readability.

2. The notation k is used too often. It's first used to denote the other of star, and then is used to denote the data partition (capitalized though), and latter it also denotes the iterations.

We thank the reviewer for the nice suggestion. We have modified the description of the elastic principal graph in the method section, now k is only used to denote the order of a star.

REVIEWERS' COMMENTS:

Reviewer #1 (Remarks to the Author):

The authors have addressed all comments and have significantly improved the method and the paper.

Reviewer #2 (Remarks to the Author):

The authors have addressed most of my comments and substantially updated the manuscript. In particular, they systematically compared STREAM with the current major lineage algorithms in the revised version and add some novel features in addition to applying EIPiGraph to this new field. I have some additional comments that authors may consider to revise in order to make it more useful to the single cell research community.

1. Can authors comment on the limitations of existing algorithms and also STREAM? E.g. computation, maximum number of cells. Any future work to further strengthen STREAM? The discussion part is a bit too brief.
2. Although it seems the proposed method works well in the example datasets, it is unclear how it works on different types of data (count or normalized) and on different platforms (e.g. 10X Genomics or C1). For example, 10X Genomics has released both scRNA-seq and scATAC-seq data in a unified output (e.g. <https://support.10xgenomics.com/single-cell-gene-expression/datasets> and <https://support.10xgenomics.com/single-cell-atac/datasets>). Can authors provide some practical recommendations and examples how to prepare the data from the CellRanger output and load them into STREAM? It will make STREAM much more useful in the research community.
3. Is computation a burden for any of the analysis? The authors should add the computational complexity in the comparison table.
4. The link <http://stream.pinellolab.org/> seems slow and sometimes could not be opened even I used different computers.

The authors have addressed most of my comments and substantially updated the manuscript. In particular, they systematically compared STREAM with the current major lineage algorithms in the revised version and add some novel features in addition to applying EIPiGraph to this new field. I have some additional comments that authors may consider to revise in order to make it more useful to the single cell research community.

We are very pleased by the reviewer's assessment and again we appreciate the time she/he has spent in reading carefully our manuscript to provide constructive feedback.

1. Can authors comment on the limitations of existing algorithms and also STREAM? E.g. computation, maximum number of cells. Any future work to further strengthen STREAM? The discussion part is a bit too brief.

We thank the reviewer for the nice suggestion. We have expanded the discussion section to discuss these points:

STREAM has been extensively tested using several published datasets from different organisms (zebrafish, mouse, human) and single-cell technologies (qPCR, scRNA-seq, scATAC-seq). Our comparison of STREAM with 10 other methods shows that it is among the top scoring methods on both synthetic and real datasets (best average ranking across several metrics and most balanced method for branching detection). However, there are still several general limitations for the current trajectory inference methods, including STREAM. First, most of the methods are still based on linear or tree-structure models even though we know that developmental processes may involve more sophisticated topologies, e.g. cyclic or disconnected graph. Therefore, further improvement is necessary to adapt STREAM to properly describing these more general structures. Although EIPiGraph, the core algorithm behind STREAM, inherently supports learning complex graphs, more work is necessary to provide intuitive visualizations to capture and represent more sophisticated topologies. Second, recent single-cell assays are now capable of profiling millions of cells, however many of the current trajectory inference methods don't scale or have been tested only on a few thousand cells (see Supplementary Table 2). Last but not least, while integrating multi-omics is crucial to accurately describe the cellular developmental landscape, so far very few trajectory inference methods (Monocle2 and STREAM) have been applied to analyze single cell chromatin accessibility. Therefore, we expect further development in this direction and on methods to properly integrate the different trajectories obtained from different omics data.

2. Although it seems the proposed method works well in the example datasets, it is unclear how it works on different types of data (count or normalized) and on different platforms (e.g. 10X Genomics or C1). For example, 10X Genomics has released both scRNA-seq and scATAC-seq data in a unified output (e.g. <https://support.10xgenomics.com/single-cell-gene-expression/datasets> and <https://support.10xgenomics.com/single-cell-atac/datasets>

) Can authors provide some practical recommendations and examples how to prepare the data from the Cell Ranger output and load them into STREAM? It will make STREAM much more useful in the research community.

We thank the reviewer for raising this important point. To make STREAM easier to adopt and to provide clear guidelines, in the Supplementary Note 5, we have added the links to four tutorials illustrating different cases (and also available here <https://github.com/pinelloolab/STREAM#tutorial>).

We also agree that supporting the 10x unified output from Cell Ranger is an important feature that a modern single cell tool should have. Therefore, we have updated STREAM and described in Supplementary Note 5 how to import the Cell Ranger output.

To load and use 10x Genomics single cell RNA-seq data processed with Cell Ranger it is necessary to set the parameter 'file_format' to 'mtx', e.g.:

```
adata=st.read(file_name='./filtered_gene_bc_matrices/hg19/matrix.mtx',
file_format='mtx')
```

To load and use 10x Genomics single cell ATAC-seq data processed with Cell Ranger, as in the case of the scRNA-seq, it is necessary to set the parameter 'file_format' to 'mtx', e.g.:

```
import stream_atac

adata =
stream_atac.preprocess_atac(file_count='./filtered_peak_bc_matrix/matrix.mtx',

file_sample='./filtered_peak_bc_matrix/barcodes.tsv',
file_region='./filtered_peak_bc_matrix/peaks.bed',file_format='mtx')
```

3. Is computation a burden for any of the analysis? The authors should add the computational complexity in the comparison table.

This is an excellent suggestion and we reasoned on what would be best way to assess the computational complexity of each method. Unfortunately, many of these methods we have compared don't provide sufficient information on the computational complexity. In addition, it is hard for us to infer their complexity just from their implementation since each method is composed of several steps and different methods are implemented with different programming languages/libraries. However, we reasoned that we can summarize in a table, for the different methods, two important points related to the computational complexity for practical applications:

- 1) Datasets analyzed in the original study with details on the number of cells, technology used and organism.
- 2) Execution time for all the methods based on the benchmark datasets we have used for all the comparisons and on the same hardware.

4. The link <http://stream.pinelloab.org/> seems slow and sometimes could not be opened even I used different computers.

Currently the website is hosted in a small lab machine so when multiple users try to run their analyses, the website may not be responsive. To speed up the analyses and to provide a more stable, fast and long-term solution, we propose now two different workflows:

- 1) We provide a Docker image so the users can easily deploy the current website to local or cloud machines. Clear instructions are available on the new dedicated GitHub page: https://github.com/pinelloab/STREAM_web and in Supplementary Note 4.
- 2) When large-scale or more advanced analyses are required we suggest using first the bioconda package 'stream' instead of the STREAM interactive website. This website was designed as a simplified version of the bioconda package for users without computational skills. Using the bioconda package each step of the trajectory inference can be tweaked and customized if necessary. Once the user is satisfied with the results it is possible to save a summary report with the function `save_web_report ()`. This function creates a .zip file that can be imported and visualized in the website under the section 'Visualize'. We are documenting this new workflow in Supplementary Note 5.

We believe the two suggested workflows allow users to easily run STREAM on private machines, perform advanced analysis, and quickly compute, visualize and share trajectory results.